



Contrasting stable water isotope signals from convective and large-scale precipitation
phases of a heavy precipitation event in Southern Italy during HyMeX IOP 13
Keun-Ok Lee[1], Franziska Aemisegger[2], Stephan Pfahl[2,3], Cyrille Flamant[4], Jean-Lionel Lacour[5],
and Jean-Pierre Chaboureau[1]
[1]Laboratoire d'Aérologie, Université de Toulouse, CNRS, UPS, Toulouse, France
[2]Institute for Atmospheric and Climate Science, ETH Zurich, 8092 Zurich, Switzerland
[3]Institute of Meteorology, Freie Universität Berlin, Berlin, Germany
[4]LATMOS/IPSL, CNRS, Sorbonne Université and Université Paris-Saclay, Paris, France
[5]Institute of Earth Sciences, University of Iceland, Reykjavik, Iceland

14                                      ABSTRACT

The dynamical context and moisture transport pathways associated with a heavy precipitation event (HPE) in
Southern Italy (SI) are investigated with the help of stable water isotopes (SWIs). The event occurred during
the intensive observation period (IOP) 13 of the field campaign of the Hydrological Cycle in the Mediterranean
Experiment (HyMeX) on 15 and 16 October 2012. SI experienced intense rainfall of 62.4 mm over 27 hr with
two precipitation phases during this event. The first one (P1) was induced by convective precipitation linked
to a frontal feature, while the second one (P2) was mainly associated with precipitation induced by large-scale
uplift. The moisture transport and processes responsible for the HPE are analysed using a simulation with the
isotope-enabled regional numerical model COSMO$_{\text{iso}}$. Backward trajectory analyses based on this simulation
show that the air parcels arriving in SI during P1 originate from the North Atlantic, and descend within an
upper-level trough over the north-western Mediterranean. The descending air parcels reach elevations below
1 km over the sea and bring dry and isotopically depleted air (median $\delta^{18}O \leq -25$ ‰, water vapour mixing
ratio $q \leq 2$ g kg$^{-1}$) close to the surface, which induces strong surface evaporation. These air parcels are rapidly
enriched in SWI ($\delta^{18}O \geq -14$ ‰) and moistened ($q \geq 8$ g kg$^{-1}$) over the Tyrrhenian Sea by taking up moisture
from surface evaporation and potentially from evaporation of frontal precipitation. Thereafter, the SWI-



enriched low-level air masses arriving upstream of SI are convectively pumped to higher altitudes, and the
SWI-depleted moisture from higher levels is transported towards the surface within the downdrafts ahead of
the cold front over SI, producing a large amount of precipitation of convective nature in SI. Most of the moist
processes (i.e. evaporation, convective mixing) related to the HPE take place during the 18 hours preceding
the occurrence of P1 over SI. Four hours later, during the second precipitation phase P2, the air parcels arriving
over SI mainly originate from North Africa. The strong cyclonic flow around the eastward moving upper-level
trough induces the advection of a moist and SWI-enriched African plume towards SI, and leads to large-scale
uplift of the warm African air mass along the cold front. This brings moist and SWI-enriched air masses
(median $\delta^{18}O \geq -18$ ‰, median $q \geq 6$ g kg$^{-1}$) to higher altitudes and leads to gradual rain out of the air parcels
over Italy. Large-scale ascent in the warm sector ahead of the cold front takes place during the 72 hours
preceding P2 in SI. This work sheds light on the variety of thermodynamic mechanisms occurring at the meso-
and synoptic scales and leading to two distinct precipitation phases of a HPE over the densely populated SI
region.
**1. Introduction**
The Mediterranean basin is frequently affected by deep convection resulting in heavy precipitation and
potentially leading to devastating flash floods. Deep convection generally results from complex multi-scale
interactions between large-scale, mesoscale, and microphysical processes. In the north-western Mediterranean,
the large-scale patterns associated with heavy precipitation events (HPEs) have been shown to be connected
to upper-level troughs, responsible for generating low-level northward flow of marine air masses characterized
by high values of equivalent potential temperature and precipitable water (Lin et al., 2001; Martius et al., 2006;
Nuissier et al., 2008, 2011; Ricard et al., 2012; Barthlott and Davolio, 2015). In this favourable large-scale
situation, mesoscale deep convection can occur and often produces high-impact events, along with rainfall
amounts larger than 100 mm in less than 6 hours. The origin of the moisture content feeding the deep
convective systems is an important question that has been addressed using different techniques and tools, such
as trajectory and numerical tracer analyses (*e.g.* Turato et al. 2004; Winschall et al., 2012; Duffourg and
Ducrocq, 2013; Winschall et al., 2014; Duffourg et al., 2018; Lee et al., 2018). These studies found substantial
contributions of subtropical and tropical moisture coming either from Africa (latitude $\geq 20°N$) or from the
extratropical remnants of Atlantic tropical cyclones. More recent studies (*e.g.* Lee et al. 2016 and 2017) pointed





out the intrusion of large amounts of water vapour from North Africa in the mid-troposphere (3-5 km above
sea level, ASL) feeding the deep convective systems together with the local water vapour sources over the
Mediterranean in the lower troposphere (below 2 km ASL). Moreover, the importance of intensified
evaporation over the Mediterranean Sea surface for HPE has been studied (Duffourg and Ducrocq, 2013;
Winschall et al., 2014). The vertical distribution of moisture in the atmosphere is shaped by source, transport,
and sink processes, e.g. evaporation and condensation, horizontal and vertical advection, as well as turbulent
and convective mixing.

8        To improve our understanding of the water vapour transport upstream of HPEs and the moisture cycling

during such events, humidity observations based on measurements of the most abundant stable water isotopes
(SWI) $H_2^{16}O$ alone can be limited. In this context, the observation of other, less abundant SWIs, i.e. $H_2^{18}O$ and
$HD^{16}O$ can provide relevant additional constrains (Noone et al., 2012; Pfahl et al., 2012; Aemisegger et al.
2015; Galewsky et al., 2016; Sodemann et al., 2017). Heavy and light isotopes of the water molecule are
partitioned in a very specific way during phase transitions, leading to an enrichment of the heavier molecules
compared to the lighter ones in the phase with the stronger bonds (liquid or ice) and a depletion in the other
phase (vapour). Therefore, they can provide a record of evaporation and condensation processes during the
transport of air parcels. Since the strength of fractionation depends on the meteorological conditions
(temperature and the level of saturation), SWI are a powerful indicator of phase change conditions in the
atmosphere that occur during the transport of air parcels on a broad range of scales, reflecting evaporation,
condensation, and air mass mixing processes (e.g., Sodemann et al., 2017). For instance, low $\delta^2H$ or $\delta^{18}O$
values in atmospheric water vapour ($\delta^{18}O = (Rs/RVSMOW - 1) \times 1000$, where $Rs = [H_2^{18}O]/[H_2^{16}O]$ is the
isotope ratio of a water sample and RVSMOW is the isotope ratio of the Vienna Standard Mean Ocean Water)
indicate the origin of low air mass temperatures and strong rainout of air parcels (e.g. Jacob and Sonntag, 1991;
Yoshimura et al., 2010), whereas high $\delta^2H$ or $\delta^{18}O$ indicate high air mass temperatures and recent admixture
of fresh ocean evaporate.

25        In the past, some of the most prominent applications of SWIs have been in a paleoclimate context to

infer past temperatures and moisture sources from natural archives, for groundwater studies, and in studies
investigating the water vapour budget in the stratosphere (Sherwood and Dessler, 2000; Vimeux et al., 2001;
Dessler and Sherwood, 2003; Jouzel et al., 2005). The process-based insight provided by the isotope
composition of atmospheric water, have more recently been extended to synoptic and sub-diurnal timescales,



and to the lower troposphere, where most atmospheric water vapour resides. Thanks to a tremendous expansion
in the number of datasets of water vapour isotopic composition and a substantially improved set of theories
and models for interpreting them, the related studies have been expanded during the past several years (*e.g.*
Pfahl et al. 2008; Steen-Larsen et al. 2014; Bonne et al. 2014; Aemisegger et al. 2015; Dütsch et al., 2017;
Lacour et al., 2017; Christner et al., 2018).

6          Recent studies have shown the unique information about meteorological processes registered in SWI

data. For instance, using ground-based SWI measurements and numerical simulations, Pfahl et al. (2012) and
Aemisegger et al. (2015) investigated the mixing processes of different air masses, as well as isotope
fractionation and equilibration in relationship with precipitation evaporation, during the passage of cold fronts.
Aemisegger and Papritz (2018) and Aemisegger and Sjolte (2018) showed that the important moisture uptake
by cold and dry airstreams during events of strong large-scale ocean evaporation carries a distinct SWI-
signature in water vapour. Recent studies (Schneider et al. 2016; Lacour et al. 2017) analysed the influence of
the Saharan heat low on the isotopic budget of the free troposphere offshore of West Africa on various temporal
and spatial scales, highlighting the importance of the Saharan heat low dynamics on the moistening and the
SWI enrichment of air parcels in the free troposphere over the North Atlantic. In addition, Risi et al. (2008)
used stable isotopic signal to better understand convective precipitation processes. These previous studies
evidenced the usefulness of water vapour isotope data to better understand meteorological processes and
moisture transport. Nevertheless, there are still very few studies (Risi et al., 2008 and 2010; Tremoy et al.,
2014) focusing on the application of water vapour isotopes to investigate moist processes associated with HPEs
at the mesoscale particularly in the extratropics.

21         Our study focuses on the transport of moisture associated with a HPE that occurred over southern Italy

(SI) on 15−16 October 2012 and produced precipitation over land exceeding 60 mm in 27 hr (Fig. 1a). The
HPE consists of two precipitation peaks, the first peak in the late afternoon of 15 October and the second peak
around midnight on that day. Using a combination of ground-based, airborne and space-borne observations
and numerical simulations of this HPE that occurred during the Intensive Observation Period 13 (IOP 13) of
the first Special Observing Period of the Hydrological cycle in the Mediterranean Experiment (HyMeX SOP-
1, Ducrocq et al., 2012), Lee et al. (2016) investigated the detailed dynamic and thermodynamic environments
of the two precipitation phases of the HPE. During Phase 1 (P1), rainfall was connected to convection triggered
by local low-level convergence ahead of a cold front and was favoured by moist conditions in the low levels



over the Tyrrhenian Sea. Heavy precipitation during Phase 2 (P2) was initiated over Algeria and was favoured
by the southerly flow ahead of the upper-level trough and large low-level moisture content and high sea surface
temperature in the Strait of Sicily. The penetration of the mistral wind over the Mediterranean and SI at the
end of 15 October terminated the convection activity. Thanks to the unprecedented data acquired offshore and
inland during IOP 13, the detailed moisture structure upstream of the HPE was investigated by Lee et al.
(2016). They highlighted 1) the presence of an African moisture plume favouring the efficiency of the
convection to produce more precipitation, 2) the importance of southerly flow from the warmer Mediterranean
Sea south of Sicily in enhancing the convergence ahead of the cold front, 3) the role of the upper-level trough
over southern France extending to the western Mediterranean in organizing convection at the leading edge of
the surface front. However, the moisture origin and the humidity transport pathways that are involved in the
HPE over SI have not been studied to date.
Here we investigate these moisture transport processes using trajectory calculations and SWI data
obtained from a numerical simulation with 7-km horizontal resolution. A detailed description of the data and
methodology is presented in section 2. Section 3 provides an overview of the meteorological conditions during
the two precipitation peaks related to the HPE during IOP 13. Section 4 discusses the isotope signals and relates
them to the moisture transport history. A summary and a discussion of the findings of the present study are
given in section 5.
**2. Data and method**
*2.1. COSMOiso model configuration and simulation*
The COSMO model (Steppeler et al., 2003) is a non-hydrostatic, limited-area numerical weather and climate
prediction model and is operationally used by several European weather services. The isotope implementation
(COSMOiso; Pfahl et al., 2012) is similar to other Eulerian isotope models (e.g. Joussaume et al., 1984; Sturm
et al., 2005; Blossey et al., 2010). COSMOiso has already shown its capability to simulate the variations of
stable water isotopes at the event-timescale (Pfahl et al., 2012; Aemisegger et al. 2015) as well as in a
climatological sense (Christner et al. 2018; Dütsch et al. 2018). It includes two additional parallel water cycles
for each of the heavy isotopes ($H_2^{18}O$, $HD^{16}O$), which are used purely diagnostically and do not affect other
model components. The heavy isotopes experience the same processes as the light isotope ($H_2^{16}O$), except



during phase transition, when isotopic fractionation occurs. A one-moment microphysics scheme is used and
convection is parameterised following Tiedtke (1989). For a detailed description of the physics and isotope
parameterisations, see Doms et al. (2011) and Pfahl et al. (2012), respectively.
Operational analysis data from the European Centre for Medium-Range Weather Forecasts (ECMWF)
are used as boundary and initial conditions for the standard model variables. For the period in October 2012,
these data are available every six hours with a spectral resolution of T1279 and 91 vertical levels and are
interpolated to the COSMO grid. After the model initialisation, information from the analysis data is only used
at the model boundaries, employing a relaxation scheme following Davies (1976). For the water isotopes,
initial and boundary data are taken from a historical isotope global circulation model IsoGSM simulation by
Yoshimura et al. (2008), who employed the IsoGSM global model data using a nudging technique (see also
Pfahl et al., 2012).
In this study, a horizontal grid spacing of 0.0625° (in a rotated grid), corresponding to approximately 7
km, and 40 hybrid vertical levels are used. The model domain covers the northwestern Mediterranean, the east
Atlantic, and the northern African regions (longitude ranging from –16.3 to 22.8°E and latitude ranging from
17.3 to 49.2°N). The simulation starts at 00 UTC on 12 October 2012, and runs for 5 days producing output
fields every hour.
*2.2. Trajectory calculation*
Air parcel backward trajectories (Wernli and Davies, 1997; Sprenger and Wernli, 2015) are calculated using
the three-dimensional wind fields from the COSMOiso simulation. In total 1440 trajectories per hourly time
step are started from 60 grid points within a box over SI (bounded by 15.2°W, 16.6°W, 39.6°N, 41.3°N) and
24 different vertical levels between 1000 and 400 hPa. The trajectories are computed five days back in time.
Generally, after 3 days the COSMO trajectories move out of the regional model domain. The air parcel position
as well as the interpolated conditions ($\delta^{18}O$, water vapour mixing ratio -$q$, surface evaporation) along the
trajectories are written as an output every hour. In this study, two series of trajectories, starting at the times of
the two precipitation peaks (20 UTC on 15 October 2012 and 00 UTC on 16 October 2012) over SI are
discussed.
*2.3. $q-\delta$ analysis*



As variations in δ are tied to those in humidity, $q$, the $q$−δ space is often used for the interpretation of the
information contained in δ. The theoretical framework for interpreting paired $q$−δ data is based on a set of
simple models that account for mixing and a range of condensation conditions (Noone, 2012). The isotopic
depletion of water vapour that undergoes condensation at equilibrium can be described by a Rayleigh
distillation model as $\delta = (\alpha-1) \ln (q/q_0) + \delta_0$, in which $q_0$ and $\delta_0$ are the humidity and the isotopic composition
of the water vapour source, and $\alpha$ is the coefficient of fractionation. In this study, $q_0$ and $\delta_0$ are set to 15 g kg$^{-1}$
and −10 ‰, respectively. The mixing model is $\delta = q_0 (\delta_0 - \delta_F) 1/q + \delta_F$, in which the subscript $F$ denotes the
flux into the volume of interest, here set to −12 ‰.
Mixing and distillation of water vapour from different sources can occur over a wide range of
combinations and produce $q$−δ pairs in between these two boundary models. A Rayleigh model with a tropical
water vapour source can generally be used to describe the lower limit of the domain of existence of $q$−δ pairs.
The upper limit of this domain can be described by a mixing model between depleted and dry air from the
upper troposphere and enriched and humid air from the tropical boundary layer. The large-scale distribution of
water vapour isotope ratio is conveniently viewed as a balance between the depleting effects of condensation
(such as in a Rayleigh processes), mixing of air masses with vapour of differing isotopic composition during
large-scale transport and the enriching effects during supply from a boundary layer source (Noone 2008;
Galewsky and Hurley 2010). Also note that raindrop re-evaporation can lead to $q$−δ pairs below the Rayleigh
distillation model (Worden et al., 2007).
**3. One HPE with 2 precipitation phases over southern Italy**
From 00 UTC on 15 October to 03 UTC on 16 October 2012, the SI area (box marked by 'SI' in Figure 1) was
affected by a HPE, with two phases of precipitation. The large amount of maximum precipitation (in total 62.4
mm over 27 hr) recorded by the rain gauge network (Fig.1a) is realistically reproduced by COSMOiso
simulation (maximum precipitation of 59 mm, Fig. 1b) both in terms of amplitude and spatial distribution. The
temporal evolution of the COSMOiso domain-averaged total precipitation within the SI area (bars in Figure 2)
shows a large precipitation within the SI in excess of 10 mm between 19 UTC on 15 October and 01 UTC on
16 October, with two distinct precipitation phases: 1) a convective precipitation phase (**P1**) in the late afternoon
(19−21 UTC) on 15 October (dashed line in Fig. 2), and 2) a large-scale precipitation phase (**P2**) just before
midnight (22−00 UTC) on that day (solid line). The precipitation associated with P1 is delayed by 4 hours in



the COSMOiso simulation compared to the precipitation recorded by the rain gauge network, which shows a
peak at 16−18 UTC (grey line with dot in Fig. 2), while the precipitation during P2 phase is closely reproduced
by the simulation with a reasonable timing (~1 hour early, with the measured peak occurring at 23−01 UTC).
P1 is related to rain from the convection parameterization, and P2 is related to rain associated with large-scale
vertical motion. In the following, 20 UTC on 15 October and 00 UTC on 16 October are considered as times
representative of P1 and P2, respectively, while 16 UTC on 15 October is considered as representative of the
pre-HPE conditions.

8        At 16 UTC on 15 October 2012, an upper-level trough located over south-eastern France, extends to

northern Algeria. Sea-level pressure values lower than 1002 hPa can be observed over south-eastern France
extending to northern Italy (Fig. 3a) with the associated cyclonic flow seen at 850 hPa. Strong northerly mistral
and tramontane winds associated with very cold and dry air and thus very low potential temperature, $\theta$, are
located over the Gulf of Lion ($\leq$ 302 K, dark-blue area in Fig. 3b). At the same time, a very warm and moist
air mass with high $\theta$ values ($\geq$ 330 K, red area in Fig. 3b) is transported from the northern Africa to Sicily at
850 hPa, ahead of the trough, where the south-westerly and southerly winds converge. Over the Tyrrhenian
Sea ('TY' box in Fig. 1b), upstream of SI, a large horizontal $\theta$ gradient (308−326 K) can be seen at 850 hPa,
indicating the elongation of the surface cold front along a southwest to northeast axis.

17       During the two precipitation phases at 20 UTC (Fig. 3c, d) and 00 UTC (Fig. 3e, f), the upper-level

trough and the cold front propagate towards the south-east while the warm and moist air mass coming from
tropical Africa persists upstream of SI. At 20 UTC (Fig. 3c, d), southerly winds (10−15 m s$^{-1}$) transport the
warm and moist air mass with high $\theta$ values ($\geq$ 326 K) from the Strait of Sicily to SI. The frontal wind
convergence of south-westerly and southerly winds (10−15 m s$^{-1}$) can be seen upstream of the HPE at the 850-
hPa level. Then at 00 UTC when the trough is located in the southern Tyrrhenian Sea with the low-level mistral
air mass at the north-eastern edge, strong cyclonic flow can be identified over the SI region while the warm
and moist air mass ($\theta \geq$ 328 K) over the Strait of Sicily is continuously advected towards SI (Fig. 3e). Overall
the synoptic evolution simulated by COSMOiso is similar to the one analysed using an observational dataset
by Lee et al. (2016).

27       The temporal evolution of the domain-averaged $\delta^{18}O_v$ in water vapour and $q$ within the SI area at the

first model level (approximately 20 m ASL) (Fig. 4) shows the different behaviour during IOP 13. While the
$q$ value increases gradually to 13.5 g kg$^{-1}$ until 19 UTC, just before P1, the $\delta^{18}O_v$ value maximizes to −13.6 ‰





at 16 UTC and then decreases during P1 to −15 ‰. During P2, the $\delta^{18}O_v$ value increases shortly to −14.6 ‰
whereas the $q$ value continues to decrease to 8 g kg$^{-1}$. The detailed 3-D history and structure of $\delta^{18}O$ and $q$ of
the air parcels associated with P1 and P2 over SI will be shown in the following section.
**4. SWIs distribution in the environment of the HPE**
*4.1. Distribution of SWIs over the Mediterranean prior to the HPE*
Figure 5 shows the horizontal distributions of $q$ at 850 hPa, and $\delta^{18}O_v$ at 850 and 600 hPa at 16, 20, and at 00
UTC. At 16 UTC, Fig. 5a shows two bands of large $q$ values in excess of 6 g kg$^{-1}$ at 850 hPa upstream of the
HPE, one over TY where the cold front is located (large horizontal $\theta$ gradient in Fig. 3b), and another one
across north Africa extending towards SI where the African moist plume is located ($\theta \geq 330$ K, Fig. 3b). The
two bands of large $q$ are associated with $\delta^{18}O_v$ values larger than −16 ‰ (Fig. 5b), while the mistral and the
tramontane, the low-level strong and cold northerly winds, are associated with very low $\delta^{18}O_v$ values, less than
−24 ‰. At 600 hPa (Fig. 5c), large $\delta^{18}O_v$ values in excess of −22 ‰ can be seen upstream of the SI area at the
southern edge of the surface cold front. This signature can be explained by the transport of water vapour to
higher levels by updrafts along the front. Another interesting point we can see by comparing the $q$ with the
$\delta^{18}O_v$ maps (crescent closed by dashed line, Fig. 5a−b) is that an additional band of enriched water vapour
($\delta^{18}O_v \geq -18$ ‰, Fig. 5b) is found at the southern boundary of the mistral (and the tramontane), over western
Corsica and Sardinia, in a region of still relatively low $q$ values ($\leq 5$ g kg$^{-1}$, Fig. 5a). This SWI-enriched band
reflects the moisture brought to higher levels by convective updrafts that develop within the strong mistral
outflow over the warm sea surface, typical of cold-air outbreaks. In this region, a band of moderate brightness
temperature at 10.8 μm (230−240 K, altitudes about 5−6 km) is measured by the Spinning Enhanced Visible
and Infrared Imager on board the geostationary Meteosat Second Generation satellite (not shown, see Fig. 4
of Lee et al., 2016). In the simulation, weak precipitation is also produced in this region from clouds located
mostly below 5 km above sea level (ASL) (not shown).
At 20 UTC (Fig. 5d−f), the band with large $q$ and large $\delta^{18}O_v$ values corresponding to the air mass ahead
of the cold front moves close to SI, while the African moisture plume has moved further north around the
trough. The hourly evolution of the moist and SWI-enriched air mass over the TY during the period 16−20
UTC can also be seen in the hourly evolution of $\delta^{18}O_v$ in Fig. 6, which shows the average $\delta^{18}O_v$ in 1-km deep
layers spanning from 1 to 7 km ASL in the TY region from 09 UTC on 15 October to 09 UTC on 16 October



together with the average $\theta$ values at 850 hPa within TY. From 09 UTC to 19 UTC on 15 October, while the
average $\theta$ value at 850 hPa is consistently high at 322 K, the $\delta^{18}O_v$ values between 1 and 5 km ASL slightly
increase but the $\delta^{18}O_v$ values between 5 and 7 km ASL gradually decrease revealing the arrival of the upper-
level trough (Figure 6a).
From 20 UTC on 15 October to 07 UTC on 16 October 2012, both $\theta$ and $\delta^{18}O_v$ values start to drop
dramatically with the arrival of the cold front in the TY region (Fig. 6a). At 00 UTC (Fig. 5g) over the
Tyrrhenian Sea, where the strong cold and dry cyclonic flow prevails (Fig. 3e and 3f), the mistral is evidenced
by very low $q$ values $\leq 2$ g kg$^{-1}$ (Fig. 5g) and low $\delta^{18}O_v$ values $\leq -24$ ‰ (Fig. 5h) at 850 hPa. Higher up, at
600 hPa, the trough-related, strongly SWI-depleted air masses descending from higher altitudes show $\delta^{18}O_v$
values lower than $-46$ ‰ (Fig. 5i). In contrast to the trough, the African moisture plume is associated with
large $q$ values in excess of 10 g kg$^{-1}$ at 850 hPa level extending to the SI region (Fig. 5g). As $\theta$ decreases from
322 to 300 K in TY (Fig. 6a), the $\delta^{18}O_v$ drops more rapidly at altitudes above 3 km compared to the $\delta^{18}O_v$ drop
seen in lower altitudes, where the trough-related dry airstreams are moistened by SWI-enriched fresh ocean
evaporate. The minimum $\delta^{18}O_v$ value increases lowering the altitudes to near surface, for instance, the
minimum $\delta^{18}O_v$ values of $-23$ and $-36$ ‰ are seen at 1–2 and 2–3 km ASL respectively, while values lower
than $-47$ ‰ occur at altitudes above 3 km ASL. The hourly evolution of average $\delta^{18}O_v$ in the TY region shows
the propagation of the surface front and upper-level trough at altitudes of 1−7 km ASL, and the associated
subsidence of dry and cold air. It is worth noting that the arrival timing of cold and dry air subsidence in TY,
19−20 UTC, (Fig. 6a) corresponds to the onset of precipitation in SI, 19 UTC (vertical bars, Fig. 2).
*4.2. SWI distribution during the convective phase of precipitation*
*4.2.1. History of air parcels and related SWI evolution*
This section aims to investigate the history of the air masses involved in the convective precipitation phase P1.
Figure 7 displays the history of air parcel arriving at SI in the layer 800−700 hPa at 20 UTC on 15 October
2012. The 3-day backward trajectories shown in Fig. 7 indicate that the air parcels arriving in the SI region in
the layer between 800 and 700 hPa at 20 UTC on 15 October are from the North Atlantic. These air parcels
are mostly very dry along the track, with $q$ values below 5 g kg$^{-1}$ during the 3 days except for the last 18 hours
before their arrival in SI (Fig. 7a). In the period between 48 and 18 hours before their arrival in SI the air
parcels descend rapidly from altitudes of 3-5 km to below 1 km ASL over the Tyrrhenian Sea, and below 2.5



km ASL over the Strait of Sicily (Fig. 7d). This penetration of dry air from upper-levels to the surface enhances
surface evaporation, leading to a sharp increase of the $q$ as well as the $\delta^{18}O_v$ values (Fig. 7a−c). Between 18
and 6 hours before arrival in SI, the median surface evaporation rate along the trajectories doubles from 0.15
to 0.32 mm hr$^{-1}$ with a peak 12 hour before the arrival in SI (Fig. 7c). A few air parcels travel over the Strait
of Sicily towards SI where mixing with the moist and SWI-enriched moisture plume from North Africa occurs
(Fig. 5d−f). The median $q$ values along the trajectories increases by a factor 2.5 from 3.8 to 8.4 g kg$^{-1}$ with the
peak 10 hour before arrival in SI, whereas the median $\delta^{18}O_v$ value increases from −27 to −18 ‰ (not shown).

8       Figure 8 displays the $q−\delta^{18}O_v$ scatter diagram along the entire trajectories seen in Fig. 7 at different times

before their arrival at SI. Figure 8 also shows that the $q$ and $\delta^{18}O_v$ values rapidly increase in the last 12 hours
prior to arrival in SI. Between 60 and 12 hours before the arrival of the air parcels in SI (Fig. 8a, b), the $q$ and
$\delta^{18}O_v$ values are still relatively small, i.e. average $q$ of 2−6 g kg$^{-1}$, average $\delta^{18}O_v$ between −25 and −19 ‰, in
the dry pocket of the upper-level trough. During the last 12 hours (black star, Fig. 8b), the average $q$ is about
9 g kg$^{-1}$, and the average $\delta^{18}O_v$ is about −17 ‰. During this time, the $q−\delta^{18}O_v$ evolution follows a curve that
lies close to a typical Rayleigh line for the Mediterranean condition (SST of 26°C, blue line), indicating the
onset of precipitation. Several points fall substantially below this Rayleigh line, suggesting a precipitation
recycling by partial re-evaporation of rain drops (Worden et al., 2007).

17       Between 6 and 3 hours before their arrival in SI, the upper to low-level trajectories (grey to purple dots

in Fig. 9a, b) follow a mixing line (orange dashed line) during their descent while the lowermost trajectories
(black and grey dots) partly follow a Rayleigh distillation line (blue line). This shows that the descending air
parcels mix with the air parcels from lower altitudes, and near surface air parcels mix between surface
evaporation and background vapour. During P1 (Fig. 9c), the $q−\delta^{18}O_v$ evolution at all levels lies on and below
the Rayleigh line, suggesting that air parcels are representative of the convective updraft after condensation of
the rain drops ($q−\delta^{18}O_v$ along the Rayleigh curve) and that some air parcels took up the evaporated moisture
from falling precipitation.
*4.2.2. Horizontal distribution of SWIs*
At 20 UTC, the precipitation feature over SI is associated with a region of enhanced convective activity and
many convection cells extending from SI to the Strait of Sicily (area closed by dashed line in Fig. 10a) ahead
of the surface cold front where westerly and north-westerly winds prevail at 542 m ASL (Fig. 10b) and the





frontal south-westerly wind is dominant at 2455 m ASL (Fig. 10c). Within the precipitation area, relatively
low $\delta^{18}O_v$ values i.e. $\leq -16$ ‰ are found at 542 m ASL while relatively high $\delta^{18}O_v$ values $\geq -19$ and $-28$ ‰
are found at 2455 m and 5565 m ASL, respectively (Fig. 10b−d), showing the signature of strong and deep
convective mixing that brings SWI-depleted moisture towards the surface within the downdrafts and SWI-
enriched moisture is pumped to higher altitudes within the updrafts. This signature is consistent with the
temporal evolution of average $\delta^{18}O_v$ in SI. Figure 6b shows a larger $\delta^{18}O_v$ increase at high altitudes of 4−7 km
ASL (green to purple lines in Fig. 6b) than at lower altitudes of 1−3 km ASL (black to yellow lines) from 19
to 22 UTC. The SWI-enriched air masses with high $\delta^{18}O_r$ values in rain ($\geq -10$ ‰), are distributed over the
TY region (Fig. 10e−f) and SWI-enriched air masses with high $\delta^{18}O_s$ values in snow ($\geq -20$ ‰), are aligned
ahead of the cold front over Sicily (Fig 10g). The depletion of water vapour and the enrichment of rain water
and snow over the TY indicate the uptaking by the air mass of evaporated moisture from falling hydrometeors.

12        At the same time, the African moisture plume is associated with SWI-enriched vapour with large $\delta^{18}O_v$

in excess of $-12$ ‰ and SWI-enriched snow with larger $\delta^{18}O_s$ value than $-12$ ‰ toward the southern tip of the
precipitating area at 5565 m (Fig. 10d, g), indicating the continuous supply of the enriched moisture plume
from North Africa to SI. We can see this constantly large $\delta^{18}O_v$ values in SI at all altitudes between 1 and 7 km
during IOP13 in Fig. 6b. The dry pocket of the upper-level trough is distinguished by SWI-depleted vapour air
masses with low $\delta^{18}O_v \leq -37$ ‰ at 2455 m and 5565 m ASL over Sardinia and Corsica (Fig. 8c, d).

18        The Lagrangian analysis indicates that most moist processes inducing precipitation during P1 take place

over a very short timescale during the last 18 hours over the Tyrrhenian Sea and the Strait of Sicily. The
descending air parcels from the mid troposphere reach altitudes below 1 km ASL at the cold front and take up
large amounts of evaporated moisture near the warm sea surface of the Tyrrhenian Sea. Then additional
moisture is taken up at altitudes below 2 km ASL from mixing with the African moisture plume that extends
from the African continent to the Strait of Sicily. During the period from 18 to 6 hour before the precipitation
peak P1, the $q$ and $\delta^{18}O_v$ values strongly increase. At the time of precipitation, strong convective mixing
processes inject the moisture that is SWI-enriched to higher altitudes and deplete near surface moisture over
SI.
*4.3. SWIs distribution during large-scale phase of precipitation*
*4.3.1. History of air parcel and related SWI evolution*





The 3-day backward trajectories in Fig. 11 evidence that the air parcels arriving at SI in the layer between 800
and 700 hPa at 000 UTC on 16 October come from North Africa and partly over the southern Iberian Plateau.
Figure 11a shows that the air parcels are consistently moist along the tracks, with $q$ values mostly $\geq 5$ g kg$^{-1}$,
in contrast to the air parcels involved in the P1 phase (see section 4.2.1). During the 3 days prior to their arrival
in SI, the air parcels are enriched with SWI, showing large $\delta^{18}O_v$ values in excess of $-24$ ‰. The air parcels
are located at low altitudes mostly below 2 km ASL (Fig. 11b, d). They continuously take up water vapour
over North Algeria and in the Strait of Sicily (Fig. 11a−b), for instance the median $q$ increases from 6.5 to 9 g
kg$^{-1}$ and the median $\delta^{18}O_v$ increases from $-18$ to $-16$ ‰ in the period from 72 to 10 hours before the
precipitation onset. The air parcels arriving at SI at 00 UTC at higher levels between 700 and 500 hPa are also
moist and SWI-enriched, originate from North Africa (not shown), and are related to the moist tropical plume.

11        These moist and SWI-enriched air parcels are also evident from the scatter diagram of $q$ and $\delta^{18}O_v$.

Figure 12 shows the relatively large $q$ and $\delta^{18}O_v$ values during the 3 days prior to their arrival in SI, i.e. $q$ of
$5-16$ g kg$^{-1}$ (average of $8-10$ g kg$^{-1}$), and $\delta^{18}O_v$ between $-12$ and $-25$ ‰ (average in $-16$ and $-18$ ‰). During
this period, the minimum $\delta^{18}O_v$ of the air parcel gradually increases from $-33$ to $-27$ ‰. In particular the moist
branch of this $q-\delta^{18}O_v$ distribution lies close to Rayleigh distillation curve (blue solid line, Fig. 12a−c) for all
3 days, indicating sustained cloud and precipitation formation. As above, values below this Rayleigh curve
point to the importance of precipitation recycling, which also occurs repeatedly during the 3-day period.
*4.3.2. Horizontal distribution of SWIs*
At 00 UTC on 16 Oct. during P2, stronger precipitation than that of P1 is produced, and the precipitation cell
is located mainly over SI (marked area closed by dashed line in Fig. 13a) where strong cyclonic south-westerly
flow $\geq 25$ m s$^{-1}$ is dominant at 2455 m and 5565 m ASL (Fig. 13c−d). Within the precipitating area, water
vapour is gradually depleted and $\delta^{18}O_v$ values are relatively low from near the surface (542 m ASL) to mid
altitudes of 5565 m ASL (Fig. 13b−d). A strong depletion of isotopes in rain water is seen at 2455 m ASL (Fig.
13f). This is due to the steady large-scale ascent of air parcels in front of the trough that lead to cloud formation
and rain out. The strong depletion of vapour in lower to mid altitudes is also evident from Fig. 6b, which shows
decreasing $\delta^{18}O_v$ values from 23 to 01 UTC (black to purple lines). It is worth noting that $\theta$ increases
continuously until 23 UTC and reaches 327 K in SI (thin line with white circles in Fig. 6b), while $\theta$ is rather
constant before the arrival of the front and trough and the peak value is about 5 K lower in TY compared to SI



(thin line with white circles in Fig. 6a, b). This reflects the influence of the moisture plume from North Africa.
The moist and enriched African moisture plume including high $\delta^{18}O_v$ of vapour in excess of −26 ‰ is advected
by the strong south-westerly flow from the Strait of Sicily to SI (Fig. 13c−d). This is consistent with the rapid
re-enrichment of vapour in SI after the precipitation (Fig. 6b). Then after 04 UTC, with the arrival of the front
and upper-level trough, the vapour becomes more depleted at all levels (Fig. 6b).

6           The Lagrangian analysis indicates that the moistures that feeds the convection during P2 is coming from

air parcels that bring moisture from North Africa and take up additional moisture (2−3 g kg$^{-1}$) over the
Mediterranean. These air parcels carry moist and SWI-enriched air at layers below 2 km ASL. With the arrival
of the upper-level trough over the southern Tyrrhenian Sea, strong cyclonic flow leads to the entrainment of
air from the African moist plume to SI. During P2, the gradual depletion of water vapour takes a place at SI at
all levels.
**5. Conclusion**
During IOP 13 (15 to 16 October 2012) of the HyMeX SOP-1, SI experiences a HPE (total precipitation of
62.4 mm) with two phases of precipitation. The first one (P1) is induced by moist convective, while the second
one (P2) is mainly associated with large-scale uplift along a front. The moisture transport and processes
responsible for the HPEs that occurred over the SI area during IOP 13 have been analysed here using SWI data
obtained from a numerical simulation with COSMO$_{iso}$ at 7-km horizontal resolution. The main findings are
summarized in two schematic illustrations (Fig. 14).

20          The 3-day backward trajectory analysis shows that the air parcels arriving in SI during P1 originate from

the North Atlantic and descend within the upper-level trough over the north-western Mediterranean Sea. The
descending air parcels arriving at very low levels (below 1 km) are very dry and SWI-depleted (median $\delta^{18}O$
$\leq$ −25 ‰, water vapour mixing ratio, $q \leq 2$ g kg$^{-1}$), and rapidly take up a large amount of water vapour from
ocean evaporation (grey encapsulated area in Fig. 14a). As a consequence, it becomes enriched in SWI ($\delta^{18}O_v$
$\geq$ −14 ‰) in a very short time span over the Tyrrhenian Sea also probably by taking up evaporated moisture
from falling precipitation as hinted by the analysis of the trajectory data in the $q-\delta$ space (points falling below
the Rayleigh distillation line). Additional moisture is taken up over the Strait of Sicily at altitudes below 2 km
ASL from mixing with the enriched moisture plume from Africa ($\delta^{18}O_v \geq$ −12 ‰). The SWI-enriched low-
level air masses arriving upstream of SI are convectively pumped to higher altitudes, producing precipitation


over SI, and the SWI-depleted moisture is transported towards the surface within the downdrafts ahead of the
cold front (red and blue arrows, Fig. 14a).
During P2 (Fig. 14b), just a few hours after P1, the origin of the air parcels arriving at SI is totally
different, i.e. mostly from North Africa. The air parcels are moist and associated with large $\delta^{18}O$ values (bottom
most arrow, median $\delta^{18}O_v \geq -12$ ‰, median $q \geq 6$ g kg$^{-1}$). With the arrival of the upper-level trough ($\delta^{18}O_v \leq$
$-48$ ‰ at 600 hPa) and low-level mistral ($\delta^{18}O_v \leq -24$ ‰ at 850 hPa) over the southern Tyrrhenian Sea, the
strong cyclonic flow around the trough (grey dashed line in Fig. 14b) induces the advection of the moist plume
towards SI and leads to large-scale uplift of the warm and moist African air mass along the cold front. It brings
moisture and leads to gradual rain out of the air parcels over Italy (following Rayleigh distillation).
For the convective precipitation phase (P1), most of the moisture processes producing the HPE take
place during the last 18 hours before the arrival over SI, while the large-scale advection of SWI-enriched
moisture from the African plume by strong cyclonic flow lasts about 72 hours during the large-scale
precipitation phase (P2). In both phases, the air parcels take up substantial amount of water vapour over the
Mediterranean.
Using the hourly 3-D water vapour isotope data, we highlight the large variety of moisture sources and
transport pathways that induced the two phases of the HPE in South Italy during IOP13, and the isotopic
characteristics of various air masses associated with the upper-level trough, cold front, mistral, and African
moist plume, that were involved in convection development. We also highlight the role of the upper-level
trough over the south Tyrrhenian Sea in driving the advection of the SWI-enriched plume from North Africa
into the region of the deep convective system resulting in heavy precipitation over SI. Although our study is
entirely based on a model simulation, the results suggest that the information on mesoscale moist dynamical
processes and moisture transport that is contained in SWI, when combined with SWI observations, can provide
very useful constrains on the representation of such processes in numerical models. This will be further
investigated in future research. In addition, to further study the details of the fractionation processes in and
around deep convective systems, complementary investigations will be conducted using higher resolution
convection-permitting simulation with a 2 km grid.
**Author contribution**
KOL, FA, SP and CF planned the manuscript and analyses. SP and KOL designed the numerical simulation





and SP performed it. JLL and JPC contributed to discussion. KOL prepared the manuscript with contributions

from all co-authors.

**Acknowledgements**

This work was supported by the French Agence Nationale de la Recherche (ANR) via the IODA-MED Grant

ANR-11-BS56-0005, the MUSIC grant ANR-14-CE01-014 and the MISTRALS/HyMeX programme.

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

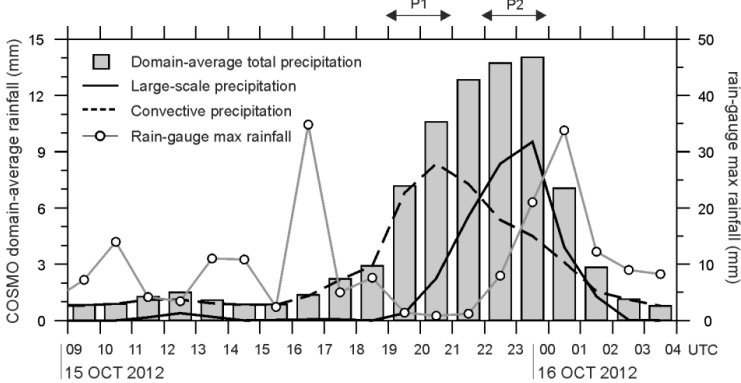

**Figure 2.** COSMOiso-produced domain-averaged total precipitation (bar), synoptic precipitation (black solid line), and
convective precipitation (dashed line) in domain of South Italy (**SI**) over the land during IOP 13. Temporal evolution of
observed maximum rainfall within the SI domain is shown by a line with dot. The location of domain SI is depicted by
the box in Figure 1.





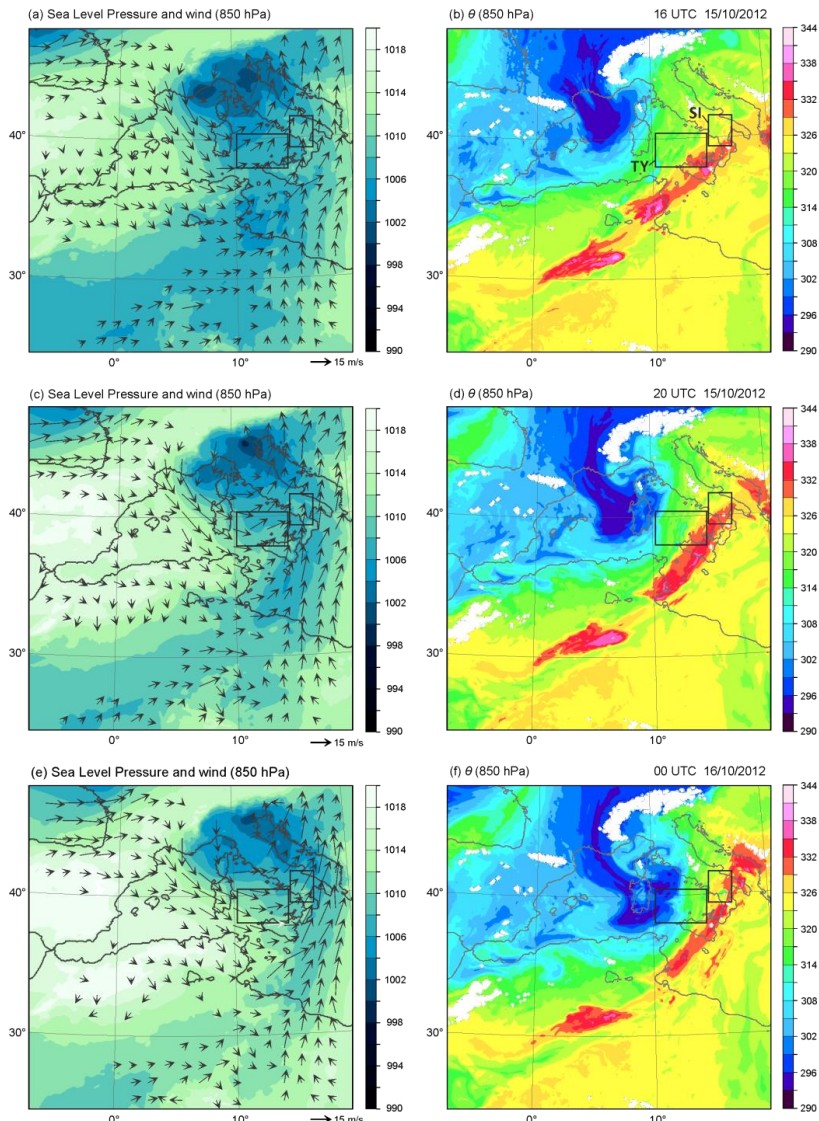

**Figure 3.** Horizontal distributions of sea level pressure (shades) and horizontal wind (black arrows) at 850 hPa (left), and

potential temperature, θ, at 850 hPa (right) at 16 UTC (top) and at 20 UTC (middle) 15 October 2012, and 00 UTC on 16

October 2012 (bottom) produced by the COSMOiso simulation. Coastal line is depicted by black line.





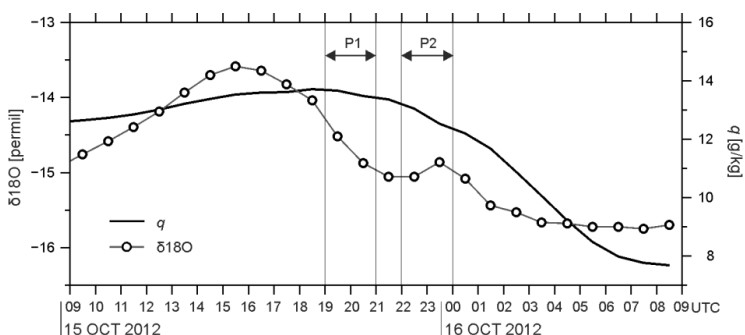

**Figure 4.** Domain-averaged $\delta^{18}O_v$ (line with dot) and $q$ (thick line) in domain of South Italy (**SI**) at the first model level

(approximately 20 m height) (limited to the grid point where the topography is lower than 20 m), from 09 UTC on 15

October 2012 to 09 UTC on 16 October 2012. The location of domain SI is depicted by the box in Figures 1 and 3.





**Figure 5.** Horizontal distributions of water vapour mixing ratio at 850 hPa (left), $\delta^{18}O_v$ at 850 hPa (middle) and $\delta^{18}O_v$ at

600 hPa (right) at 16 UTC (top) and 20 UTC (middle) on 15 October 2012, and 00 UTC on 16 October 2012 (bottom).



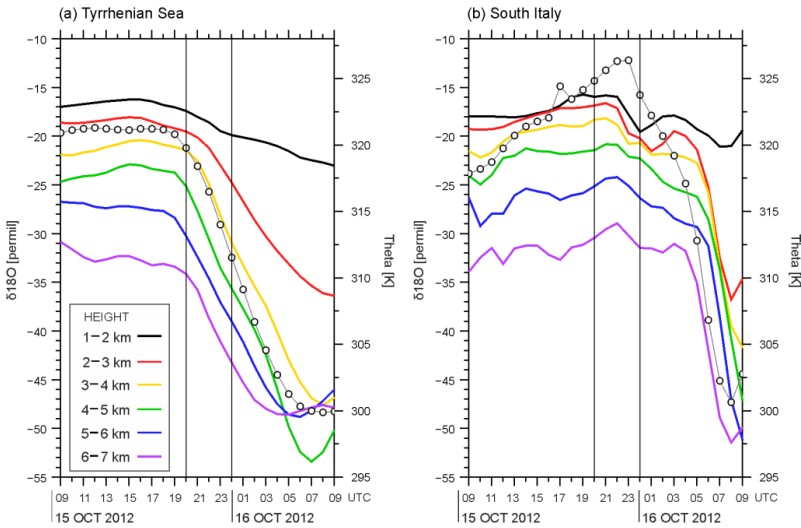

**Figure 6.** The averaged values of potential temperature ($\theta$, K) at 850 hPa (thin line with dot) and $\delta^{18}O_v$ (‰) at altitudes

of 1−2 km ASL (black), 2−3 km ASL (red), 3−4 km ASL (yellow), 4−5 km ASL (green), 5−6 km ASL (blue), 6−7 km

ASL (purple) over the sea surface upstream the HPE of IOP 13 within domains of (a) Tyrrhenian Sea (marked by 'TY' in

Figures 1, 3 and 5) and (b) South Italy (marked by 'SI') from 09 UTC on 15 October 2012 to 09 UTC on 16 October

2012.





**Figure 7.** History of air parcel arriving at **SI** in layer of 800−700 hPa at 20 UTC on 15 October 2012. (a) water vapour

mixing ratio, $q$ (g kg$^{-1}$), (b) $\delta^{18}O_v$ (‰), (c) surface evaporation (mm hr$^{-1}$), (d) altitude (km), and (e) time (hr).





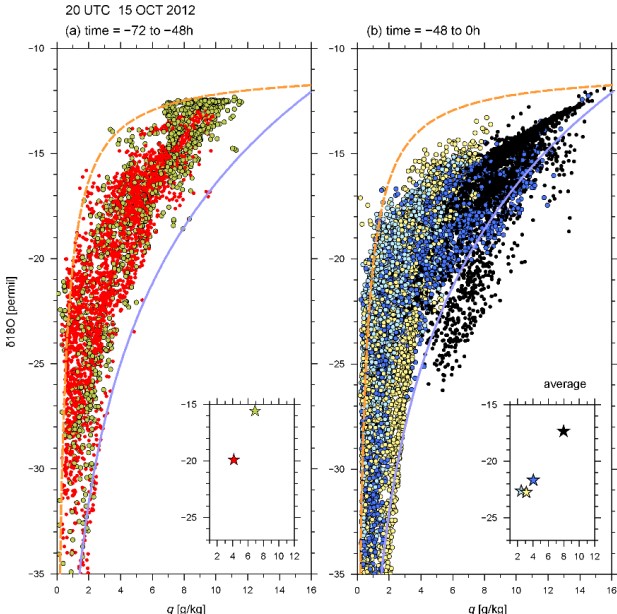

**Figure 8.** Scatter diagram of $q$ and $\delta^{18}O_v$ along the backward trajectories of Figure 7 during (a) the times between −72
and −48 hr, and (b) times between −48 and 0 hr every 12 hours from 20 UTC on 15 October 2012. The mixing and
Rayleigh lines are indicated in each panel by orange dashed line and blue solid line, respectively. The averaged $q$ and
$\delta^{18}O_v$ every 12 hours is displayed in the bottom right corner of each panel.



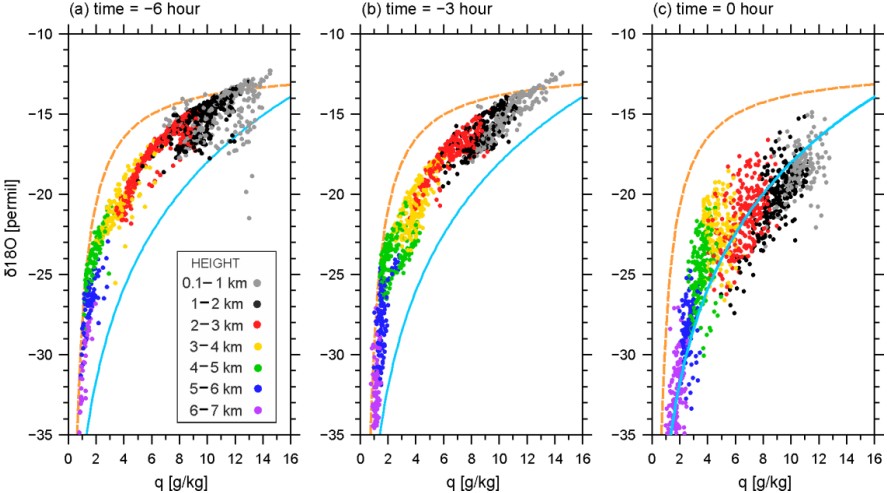

**Figure 9.** Scatter diagram of $q$ and $\delta^{18}O_v$ along the backward trajectories of Figure 7 but for all altitudes of 1−2 km (black

dots), 2−3 km (red dots), 3−4 km (yellow dots), 4−5 km (green dots), 5−6 km (blue dots), and 6−7 km (purple dots) at (a)

−6 hr, (b) −3 hr, and (c) 0 hr from 20 UTC on 15 October 2012. The mixing and Rayleigh lines are indicated in each panel

by orange dashed line and blue solid line, respectively.





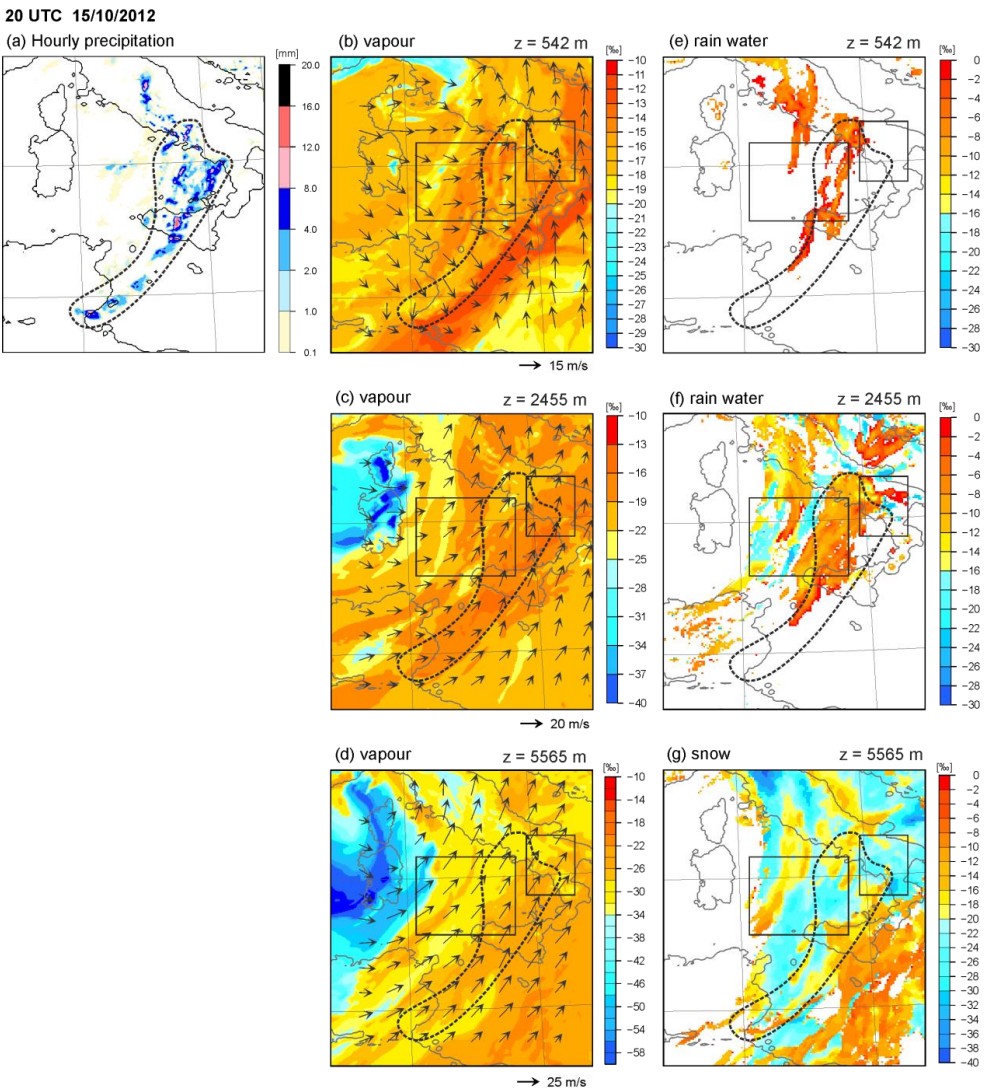

**Figure 10.** Horizontal distributions of (a) surface hourly precipitation (mm), $\delta^{18}O_v$ (‰) at (b) model level 8 (about 542 m ASL), (c) model level 16 (about 2455 m ASL), and (d) model level 23 (about 5565 m ASL, $\delta^{18}O_r$ (‰) at (e) 542 m ASL and (f) 2455 m ASL, and $\delta^{18}O_s$ (‰) at 5565 m ASL at 20 UTC on 15 October 2012. The precipitating area is marked by the area enclosed by the dashed line.





2    **Figure 11.** Same as Figure 7 but for the air parcel arriving at SI in layer of 800−700 hPa at 00 UTC on 16 October 2012.




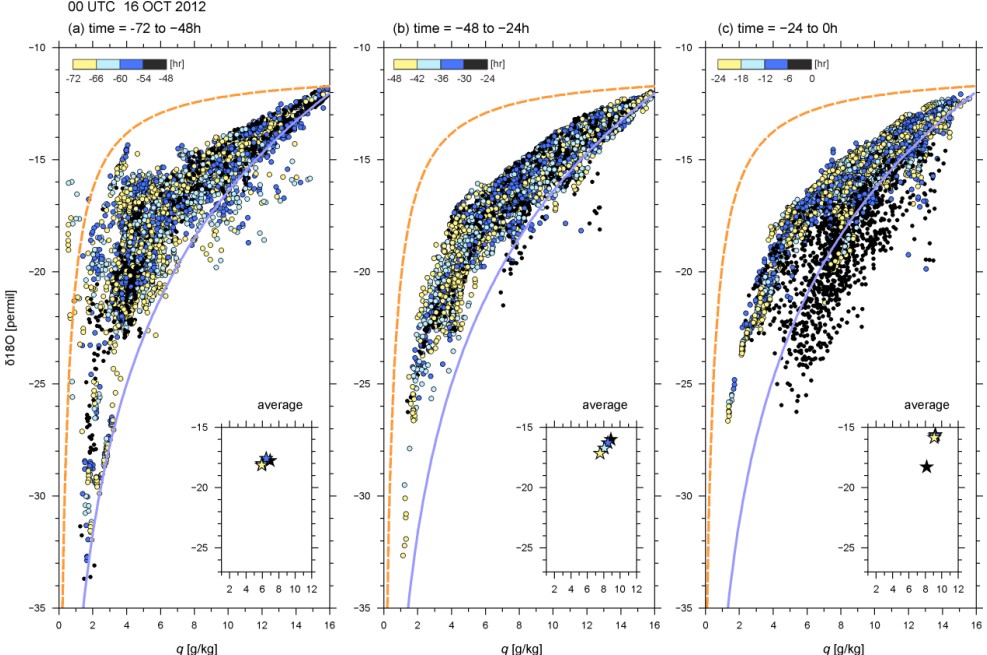

**Figure 12.** Scatter diagram of $q$ and $\delta^{18}O_v$ along the backward trajectories of Figure 11 during (a) the times between $-72$

and $-48$ hr, (b) times between $-48$ hr and $-24$ hr, and (c) times between $-24$ hr and 0 hr from 00 UTC on 16 October

2012 every 6 hours. The mixing and Rayleigh lines are indicated by orange dashed line and blue solid line, respectively.

The averaged $q$ and $\delta^{18}O_v$ every 6 hours is displayed in the bottom right corner of each panel.







2  **Figure 13.** Same as Figure 10 but for 00 UTC on 16 October 2012.





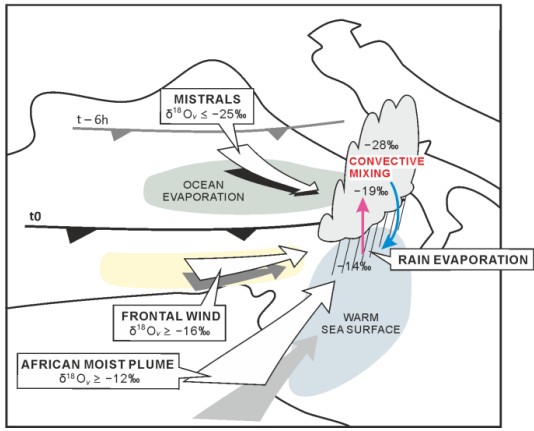
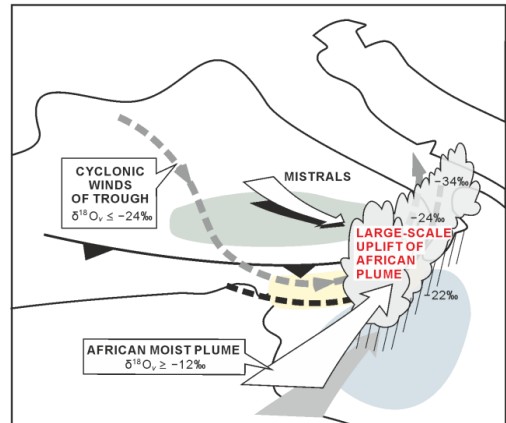

**Figure 14.** Schematics summarizing the main features of water vapour isotopologues and processes for deep convection upstream of SI and leading to the Phase 1 (a) and Phase 2 (b) of the HPE. In (a) and (b), white descending arrow indicate the mistral wind behind the edge of the cold front (thick black line). The white arrow in the yellow-shading encapsulated area illustrates the frontal wind at 850 hPa, and white arrow in the blue-shading encapsulated area (warm sea surface) indicates the elevated African moist plume. In (a), convective ascent and precipitating downdraft is depicted by red and blue arrows, respectively. In (b), the southern edge of upper trough is indicated by black dashed line and the cyclonic flow of the trough is indicated by grey dashed line.