# Peer review of "Contrasting stable water isotope signals from convective and large-scale precipitation phases of"

_Atmospheric Chemistry and Physics, 2018_

## Referee Comment (RC1) · Anonymous Referee #1 · 27 Jan 2019

General:

The authors present model simulations of isotope ratios, but there is no link to observations. Are the authors suggesting that observations form e.g. the HyMeX IOP 13 could be compared to their model results to gain additional insights? If yes, how? If not, why not just stick to a trajectory analysis? In other words, what additional insights (if any) are gained by using COSMOiso here?

It seems that the authors turn on a convection parameterization at 7 km horizontal

resolution. Are the two parameterizations of microphysical processes consistent between the deep convection parameterization and the "large scale" scale microphysics parameterization? If not, how may this affect the results regarding the isotope ratios? Are the authors sure that the model skillfully represents the partitioning between parameterized and resolved precipitation or does this partitioning depend on details of the model formulation? What might be the effects of not representing this partitioning skillfully on the simulated isotope ratios? If the results were sensitive, would it even be worthwhile to try to improve the partitioning in the model or should one just wait until deep convection parameterizations become obsolete?

I must admit that I do not fully understand the purpose of this manuscript. To me this study raises more questions than it answers. Major revisions will be necessary before I can recommend this study for publication.

Specific points:

1. Title and abstract: based on the title and abstract I would have somehow expected a connection with observations. The title should explicitly state that this is purely a modelling exercise.

2. Given that this is purely a modelling exercise, I would have expected either some sensitivity studies or else a more explicit description on how the model results presented here might be linked to either existing or to future observations.

3. Is the main point of this study a simulation of a quantity that has not been observed or are there fundamental new results that can not be found in the existing literature? If this is the case, these points should at least briefly be discussed in the light of the existing literature.

4. Why was COSMOiso used and not just a simple trajectory analysis?
* * *

---

## Referee Comment (RC2) · Anonymous Referee #2 · 1 Feb 2019

General remarks

A case study of a heavy precipitation event over southern Italy during HyMeX IOP13 in October 2012 has been discussed by far-reaching and detailed interpretation of model output. However the only validation of the results presented are numerous rain-gauge data from stations in southern Italy. Although the analysis of the model results are sophisticated and as far as possible reliable and in accordance with known synoptic and sub-synoptic flow pattern, any link to real processes is missing due to missing observations.

[Figure]

By the title the authors guide the reader to a case study of combined convective and large scale precipitation formation. But this is only partly the case. Studying the manuscript we can learn, that with the use of the selected models, how much in detail we can interpret and consequently use the COSMOiso model for small scale precipitation analysis and thus for forecasting issues. This should clearly be indicated in the title and it should be stated in the abstract that trajectory calculation and simulated stable water isotope analysis give valuable additional information for weather interpretation and forecasting purposes. The authors are asked to rephrase the title accordingly and to rewrite the abstract and conclusions. Otherwise, a case study based on model data only and without any supporting data (with the exception of the precipitation data) is not sufficient for publication.

After redirecting the scope of the paper, precipitation process information based on trajectory and SWI model output can be a helpful tool to examine actual weather situations governed by mutual evaporation-condensation processes. This is relevant scientific research and within the scope of ACP as the paper introduces additional parameters in operational weather analyses. The relevant SWI literature is cited by the authors, and the model set up is explained sufficiently. The analysis of the model results is accurate and detailed. The description of the flow field and the evaporation-condensation cycle are reliable. The structure of the paper is comprehensible. Some comments on figures, abbreviations, text etc. see "minor points".

I recommend the paper for publication in ACP after redirecting the purpose of the paper.

Minor points P 1, l 15 (new): Moisture transport pathways embedded in large scale flow and associated ... P 2, l 26: The paper "Diagnostic study of a HyMeX heavy precipitation event over Spain by investigation of moisture trajectories (Röhner, L., Nerding, K.-U., Corsmeier, U., 2016, Quarterly journal of the Royal Meteorological Society)" shows the analysis of a HPE by combination of simulations and observations including moisture trajectories. In "Sodemann et al. (2017)" the potential of airborne measurements of SWI (during HyMeX) is shown. P 3, l 15: better: ... condensation cycles

during. . . P 3, l 20f: explain the expression in brackets in a separate sentence for better readability. P 4, third paragraph: abbreviations SI, IOP, P1, P2 have been explained before in the abstract. P 6, l 21: refer to Fig. 1 P 6, l 26: refer to Fig. 2 P 8, l 13: . . . transported from northern Africa . . . P 8 l 15: isn't it 314 – 330 K instead of 308 – 326 K? P 8, l 23: isn't it western edge of the Ty-box instead of north-eastern edge? P 8, last paragraph: this paragraph should be removed to section 4. P 9, l 9: The cold front is often mentioned in the text but never indicated in the figures (with exception of Fig. 14, conceptual model). Section 4.1: The caption "Distribution of SWIs . . . prior to HPE" does not fit to the text: the interval with precipitation (16 UTC to 07 UTC) is discussed in this paragraph. P 12, l 15: Figs. 8c and 8d do not exist. P 15, l 24: . . .(grey encapsulated area in Fig. 14a). In the figure the area is called "blue encapsulated). Figure 3: use isobars for showing the surface pressure field. Indicate the position of the cold front. Figure 5: skip the upper and middle color bars. Figure 8: a color bar is missing. Figure 10: changing color bars for the sub-figures makes it difficult to interpret figure differences. Figure 13: as Fig. 10. Figure 14: the color differences (grey, blue, yellow shading) is difficult to read if manuscript is printed.

---

## Referee Comment (RC3) · Anonymous Referee #3 · 12 Feb 2019

**Review of**

*Contrasting stable water isotope signals from convective and large-scale precipitation phases of a heavy precipitation event in Southern Italy during HyMeX IOP 13*

February 2019

**Overview**

The reviewed article is a thorough case study performed using COSMOiso. While the analysis of the simulation results is highly detailed, the title and abstract of the manuscript lead the reader to expect a comparison to observations. However, the only observations used are precipitation measurements. Considering the emphasis on isotope modelling, I would have expected at least some measurements of this nature. For that reason, the purpose of the manuscript as a whole, and especially the long and detailed analysis of isotope distributions within the manuscript, is not clear.

The main question is: what additional insight is gained from the use of a model-only isotope analysis. This problem is further underlined by the presented trajectory analysis, which seems to provide the same information which is derived from the isotope concentrations. For a recommendation for publication, the authors need to better outline the purpose of the study, as well as their reasoning for using COSMOiso and its advantages compared to a trajectory analysis or even just passive tracers.

Assuming these problems can be addressed, a number of other issues remain. Below is a detailed listing of major and minor comments which should help the authors to improve their manuscript substantially.

**Major Comments**

1. Section 4 is the core of the manuscript. However, even considering that, it seems out of proportion. It is not only very long but also difficult to read and descriptive over long spans. This section would greatly benefit from being shorter and more concise.

2. Parts of the conclusion repeat contents of section 4 in too much detail, shorten and be much more concise and clear.

3. The authors need to discuss their results in a critical way which includes an explanation of the insights gained by using COSMOiso over a normal mesoscale simulation, which would be possible at a much higher resolution too. This discussion can be part of the last section *Conclusions and Discussion*.

4. Multiple figures are difficult to read, be it due to bad coloring or their size. The authors could greatly improve the manuscript's readability by making sure that figures use more contrasting color

tables, fewer contour levels and that the figure size and shape make better use of the available space. I mention specifics for certain figures throughout the minor comments, but the other figures can also be improved following those same guidelines.

5. Figures are often referenced out of order, try to keep this to an absolute minimum. This will likely require some restructuring of the text.

**Minor Comments**

| page | line(s) | comment |
|------|---------|---------|
| 3 | 1 | *large amounts of water vapor* how large? |
| 3 | 10 | *observations of the most stable water isotopes alone can be limited* this indicates that "normal" observations only look at this isotope, but I assume that the sentence refers to classical observations which simply look at the total moisture without any regard for different isotopes. This should be more clear. |
| 3 | 14, 15 | replace *in the other phase (vapor)* with just *in vapor* |
| 3 | 19 – 24 | Please specify what *high* and *low* are in this context by giving typical values |
| 3 | 29 | remove commas |
| 4 | 16 | change to *used **a** stable isotopic* or *used stable isotopic signal**s*** |
| 4 | 20 | add comma after *mesoscale* |
| 4 | 25 – 27 | move the part *that occurred (...) Mediterranean Experiment (HyMeX ...)* to a separate sentence |
| 5 | 1 – 5 | This description is difficult to follow if one is not familiar with Lee et al. (2016). This should be moved to section 3, where it can make use of Figs. 1 – 3 for a thorough but concise description of the event (see also comments on Fig. 1) |
| 5 | 3 | remove *wind* after *mistral* |
| 5 | 4 | change *convection activity* to *convective activity* |
| 5 | 8 | add *and* after the comma |
| 5 | 10 | change to *However, the origin and transport pathways of moisture have not been studied to date.* |
| 6 | 2 | specify that this is a *deep* convection scheme, since the resolution of the model has not yet been mentioned at this point |
| 6 | 3 | please add a very brief and concise description of what these physics and isotope parametrizations do, one to three sentences should suffice |
| 6 | 11 | please add some details about this model. Does it run operationally? Is it an analysis? Does it run only for specific cases? |
| 6 | 12 – 16 | Why is a resolution of 7 km used? Is this to be able to differentiate between convective and other precipitation by using the convection scheme's precipitation? Resources? Other reasons? Please specify. |

| page | line(s) | comment |
| --- | --- | --- |
| 6 | 22 | 5 day trajectories in a 5 day simulation? So just back to the start of the simulation or until they leave the domain? |
| 7 | 6 – 8 | Why are these values chosen? |
| 7 | 20 | Spell out *two* in the section title |
| 7 | 23 | hour is abbreviated with just *h* |
| Fig 1. | | Some dots have edges, others have none. Remove all edges and make sure the higher precipitation measurements are plotted on top of the lower values to keep them clearly visible and not hide maxima. The Figure is very small and the comparison is difficult to read. In Fig. 1b all contour colors from 5 to 25 mm look almost exactly the same in print, please use a color table which shows the differences more clearly. Fewer levels might help to achieve better contrast, do you really need 24 different ones? |
| 7 | 26 | *a large precipitation* rephrase |
| 7 | 24 – 27 | This sentence is too long and convoluted, simplify by moving the total precipitation amount to a separate sentence |
| 8 | 8 | remove comma after *France* |
| Fig. 3 | | Change the colored contours of MSLP to lines, chose a good interval (not too dense) and smooth the field a bit if necessary. Add colored contorus of 500 hPa geopotential to show the position of the trough. Move the vectors from the left panels to the right panes, they contain information of the same level. |
| 8 | 11, 12 | don't use *very* in scientific text, be specific |
| 8 | 13 | change *high $\theta$ values ($\geq$330 K)* to *values of $\theta \geq$330 K* |
| 8 | 16, 17 | Could convection be causing the cool areas in the 850 hPa $\theta$ map over TY? |
| 8 | 17 | It is never explained that the model does, in fact, not produce two peaks. They are only visible when separating precipitation from the convection scheme and precipitation produced by microphysics. |
| 8 | 18 | the trough is never shown, add reference to Fig. 3 after adding the 500 hPa geopotential as suggested |
| 8 | 23 | *strong cyclonic flow* there is only one arrow within the box, curvature is hard to see |
| 9 | 12, 13 | *very low* and *large* are not helpful in this context, just use the values. However, a short explanation on why the threshold between these two values is important would be helpful. |
| 9 | 26 | the front is not really close to SI |
| 10 | 27 | *mostly very dry* use values instead, be specific |
| 11 | 2 | remove *the* before *q* and $\delta^{18}O_v$, remove *values* after $\delta^{18}O_v$ |
| 11 | 6 | change to *The median q value (...) factor of 2.5* |
| Fig 7 | | Figure has lots of white space and the way the map is shown causes even more. Try to reduce this to make the important parts a bit larger. |
| 11 | 12 | replace *the average q* with *q is about 9 g kg$^{-1}$ on average* |
| 11 | 14 | It is never explained what a Rayleigh line is |

| page | line(s) | comment |
| --- | --- | --- |
| 11 | $19 - 21$ | This sentence is complicated. Also, why? |
| 11 | 28 | replace *many* with *multiple*, change *convection* to *convective* |
| 12 | 2 | usage of low/high seems inconsistent looking at the numbers |
| 12 | 13 | change to *with values of $\delta^{18}O_v$ larger than*; replace *toward* with *around* |
| 12 | $18 - 19$ | rewrite sentence |
| 12 | 25 | change to *convective mixing injects SWI-enriched moisture into higher altitudes* |
| Fig. 8 | | explain colors in the caption, some dots have edges and others don't, figure is small, you could change the aspect ratio to fit the page width for better readability. This also applies to the other figures of this type (9 and 12). |
| Fig. 10 | | panel titles say *vapour* and *rain water*, change them to clearly indicate that they show $\delta^{18}O_v$ **for** vapour/rain water. Also, model levels are not at a constant height. Does this have any effect over mountains? if so, expalain which one? Better alternative: plots for certain altitudes above sea level, e.g. 500, 2500, and 5000 m, instead of model levels. |
| 13 | 2 | replace *over* with *from* |
| 13 | 3 | Threshold of 5 g kg$^{-1}$ is not visible in the figure |
| 13 | 7 | replace *for instance* with *and* |
| 13 | 7, 8 | Use a non-breaking space between multiple units, in LaTeX to avoid line breaks between them. This can be done by using ~ instead of a space like this: 9~g~kg or in MS Word by using Ctrl + Shift + Space |
| 13 | $20 - 21$ | rephrase, also, do not use *precipitation cell* unless explicitly referring to a single convective cell |
| 13 | 24 | The depletion is hardly visible at 5500 m. |
| 14 | $1 - 5$ | going back to earlier Figures is tedious and disrupting, try to avoid if possible by restructuring the text |
| 14 | $6 - 8$ | rephrase sentence |
| 14 | 9 | replace *entrainment* with *mixing*, entrainment is usually used in the context of convective updrafts |
| 14 | 15 | replace *convective* with *convection* |
| 14 | 20 | the three paragraphs starting here are especially long and too descriptive, be more concise. Do not simply repeat details from previous sections in the conclusions. |
| 15 | 4, 5 | do not use formulations like *totally different* in scientific texts |

---

## Author Comment (AC1) · 15 Mar 2019

**Contrasting stable water isotope signals from convective and large-scale precipitation phases of a heavy precipitation event in Southern Italy during HyMeX IOP13**

**By K. O. Lee et al.**

Reply to the referees' comments

In the following, the comments made by the referees appear in black, while our replies are in red, and the proposed modified text in the typescript is in blue.

Referee #1 comments

**General Comments**

The authors present model simulations of isotope ratios, but there is no link to observations. Are the authors suggesting that observations from e.g. the HyMeX IOP 13 could be compared to their model results to gain additional insights? If yes, how? If not, why not just stick to a trajectory analysis? In other words, what additional insights (if any) are gained by using COSMOiso here?

In this study, a trajectory analysis based on a COSMOiso simulation has been done to investigate the moist processes in air masses associated with a heavy precipitation event (HPE) along their pathway. Stable water isotopes (SWI) experience fractionation during phase transition of hydrometeors, and hence can record information about evaporation and condensation processes during the transport of air parcels. Since the strength of fractionation depends on the meteorological conditions (temperature and saturation level), the SWI characteristics thus have led to an improved understanding of key processes affecting atmospheric humidity. Our study serves as a model-based proof of concept that such additional insights can also be obtained regarding convective precipitation events in the Mediterranean. In this way, we provide a basic motivation for future observational studies using SWI. To date, not many campaign datasets exist that would allow such observational studies yet, but with the recent advent of field deployable high precision laser spectrometric instruments, tailored field experiments become possible and will be done in the near future.

The analysis using three dimensional (3D) SWI (both $H_2^{16}O$ and $H_2^{18}O$) fields obtained from COSMOiso shown in Figures 5, 7, 8, 9, 10, 11, 12, and 13, give insights into the different moist processes occurring in the air parcels approaching Southern Italy. Especially the 3D $q$–$\delta$ analysis (shown in Figures 8, 9, and 12) reveal the importance of mixing, condensation, and enriching processes occurring along the moisture transport pathway. Previous studies demonstrated the usefulness of SWI to better understand meteorological processes and moisture transport, nevertheless there are still very few studies focusing on the application of SWI to investigate moist processes associated with HPEs. In particular, we see a great potential in the use of SWI for better understanding the moisture dynamics in HPEs occurring in the Mediterranean where deep convective systems are frequently observed and the origins of the moisture feeding the convection are diverse. The additional insights we can get from the COSMOiso simulation are now being more clearly emphasized in the conclusion section.

To address this, the following changes have been made.

♣ Page 16, line 1-14

"[…] We also highlight the role of the upper-level trough over the south Tyrrhenian Sea in driving the advection of the SWI-enriched plume from North Africa into the region of the deep convective system resulting in heavy precipitation over SI. Moreover, we demonstrate the importance of various moist processes such as mixing, condensation, and re-evaporation along the pathway based on the $q$–$\delta$ analysis using 3D SWI fields. Although our study is entirely based on a model simulation, the results suggest that the information on mesoscale moist dynamical processes and moisture transport that is contained in SWI, when combined with SWI observations, can provide very useful constraints on the representation of such processes in numerical models.

Our study is the first study to investigate the potential benefit of SWI in the context of a HPE in the Mediterranean. As such, our study provides a proof of concept of the usefulness of SWI data to understand the variety of origins and moist processes associated with air masses feeding the convection over SI. This will be further investigated in future research using SWI measurements obtained from various platforms, e.g. ground-based, near surface, airborne (Sodemann et al., 2017), and space-borne. Our modelling study will also allow designing forthcoming tailored field campaigns in the Mediterranean region."

It seems that the authors turn on a convection parameterization at 7 km horizontal resolution. Are the two parameterizations of microphysical processes consistent between the deep convection parameterization and the "large scale" microphysics parameterization? If not, how may this affect the results regarding the isotope ratios? Are the authors sure that the model skillfully represents the partitioning between parameterized and resolved precipitation or does this partitioning depend on details of the model formulation? What might be the effects of not representing this partitioning skillfully on the simulated isotope ratios? If the results were sensitive, would it even be worthwhile to try to improve the partitioning in the model or should one just wait until deep convection parameterizations become obsolete?

Of course the partitioning between convective and large-scale precipitation is somewhat artificial (there is not such separation in observations) and depends on the model as well as on the resolution (when going to finer resolutions, the model explicitly resolves a larger portion of the mesoscale convective activity). However, this partitioning is not the main aspect/goal of our study. In addition, we are confident that the current model setup is well-suited for the purpose of exploring the worth of isotope data for understanding mesoscale moist processes. The grid spacing of 7 km together with the Tiedtke convection scheme were used for operational predictions at the German Weather Service DWD for a long time, such that the model is very well tuned in this configuration. We also chose this relatively coarse resolution because it allows for a large model domain that reduces the dependence on the much coarser isotope boundary data (spectral resolution of T62 in IsoGSM) and enables us to calculate backward trajectories consistently over longer periods.

I must admit that I do not fully understand the purpose of this manuscript. To me this study raises more questions than it answers. Major revisions will be necessary before I can recommend this study for publication.

We appreciate the time and effort you put in this review as well your mindful comments on our paper. In this study, using the hourly 3-D water vapour isotope data, we highlight the large variety of moisture sources and transport pathways that induced the two precipitation phases, and the isotopic characteristics of various air masses associated with upper-level trough, cold front, mistral, and African moist plume, that were involved in convection development. Although our study is entirely based on a model simulation, the results suggest that the information on mesoscale moist dynamical processes and moisture transport that is contained in SWI, when combined with SWI observations, can provide very useful constraints on the representation of such processes in numerical models. As such, our study provides a proof of concept of the usefulness of SWI data to understand moist processes associated with a Mediterranean heavy precipitation event with diverse origins and pathways of moisture feeding the convection. We have worked hard to comply with all of them. Replies to each comment are listed below.

**Specific comment**

1. Title and abstract: based on the title and abstract I would have somehow expected a connection with observations. The title should explicitly state that this is purely a modelling exercise.

Agreed. We have modified the title and abstract accordingly.

♣ Page 1, line 2-3 (title)

"Contrasting stable water isotope signals from convective and large-scale precipitation phases of a heavy precipitation event in Southern Italy during HyMeX IOP 13: a modelling perspective"

♣ Page 1, line 15-17 (abstract)

"The dynamical context and moisture transport pathways embedded in large scale flow and associated with a heavy precipitation event (HPE) in Southern Italy (SI) are investigated with the help of stable water isotopes (SWIs) based on a purely numerical framework."

2. Given that this is purely a modelling exercise, I would have expected either some sensitivity studies or else a more explicit description on how the model results presented here might be linked to either existing or to future observations.

Agreed. Although our study is entirely based on a model simulation, the results suggest that the information on mesoscale moist dynamical processes and moisture transport that is contained in SWI, when combined with SWI observations from various platforms (also our response to the first general comment), can provide useful constrains on the representation of such processes in numerical models. In fact, several field campaigns involving SWI measurements are planned in the Mediterranean region.

♣ Page 16, line 9-14

"Our study is the first study to investigate the potential benefit of SWI in the context of a HPE in the Mediterranean. As such, our study provides a proof of concept of the usefulness of SWI data to understand the variety of origins and moist processes associated with air masses feeding the convection over SI. This will be further investigated in future research using SWI measurements obtained from various platforms, e.g. ground-based, near surface, airborne (Sodemann et al., 2017), and space-borne. Our modelling study will also allow designing forthcoming tailored field campaigns in the Mediterranean region."

3. Is the main point of this study a simulation of a quantity that has not been observed or are there fundamental new results that cannot be found in the existing literature? If this is the case, these points should at least briefly be discussed in the light of the existing literature.

Current SWI measurements are mainly obtained from space-borne retrievals (e.g. Schneider et al. 2017, Lacour et al., 2017) and ground-based in-situ laser spectroscopy (e.g. Aemisegger et al. 2012) as well as from monthly precipitation samples at various stations (GNIP; IAEA, 2006). The space-borne measurements provide continuous datasets in space at the global scale with coarse vertical resolution and limited precision. On the other hand, ground-based measurements with high temporal resolution are only available from a few locations and from dedicated field campaigns. In particular, the data availability for the Mediterranean region is very limited. A notable exception is the airborne dataset acquired around Corsica during the HyMeX SOP1 (Sodemann et al., 2017). However, it does not include SWI observations for the days under scrutiny in this paper. Due to these limitations, we use a model to demonstrate the usefulness of SWI data for understanding moist processes associated with a Mediterranean HPE. In this way, we provide a motivation and justification for future measurement efforts dedicated to this topic. As mentioned above, we have extended the conclusion section to make this clearer.

♣ Page 4, from line 23 (introduction)

   "SWI measurements are mainly obtained from space-borne retrievals (e.g. Schneider et al., 2016; Lacour et al., 2017) and ground-based in-situ laser spectroscopy (e.g. Aemisegger et al., 2012). The space-borne measurements provide continuous datasets in space at the global scale with coarse vertical resolution and limited precision. On the other hand, ground-based measurements with high temporal resolution are only available from a few locations and from dedicated field campaigns. In particular, the data availability for the Mediterranean region is very limited. A notable exception is the airborne dataset acquired around Corsica (Sodemann et al., 2017) during the first Special Observing Period of the Hydrological cycle in the Mediterranean Experiment (HyMeX SOP-1, Ducrocq et al., 2014). However, it does not include SWI observations for the days under scrutiny in this paper. Due to these limitations we use a model to demonstrate the usefulness of SWI data for understanding moist processes associated with a Mediterranean HPE."

♣ Page 16, line 5-14 (conclusion)

"[…]Although our study is entirely based on a model simulation, the results suggest that the information on mesoscale moist dynamical processes and moisture transport that is contained in SWI, when combined with SWI observations, can provide very useful constrains on the representation of such processes in numerical models.

   Our study is the first study to investigate the potential benefit of SWI in the context of a HPE in the Mediterranean. As such, our study provides a proof of concept of the usefulness of SWI data to understand the variety of origins and moist processes associated with air masses feeding the convection over SI. This will be further investigated in future research using SWI measurements obtained from various platforms, e.g. ground-based, near surface, airborne (Sodemann et al., 2017), and space-borne. Our modelling study will also allow designing forthcoming tailored field campaigns in the Mediterranean region."

4. Why was COSMOiso used and not just a simple trajectory analysis?
The additional insights we can get from COSMOiso simulation has been emphasized in conclusion section, as explained in the answer to the first general comment. The added value of an isotope enabled numerical simulation of this HPE event compared to a purely trajectory based study on moist processes, is that mixing, convective and cloud processes are included in a more detailed way. Trajectory-based studies have been done to investigate the contribution of different evaporative moisture sources for precipitation events (e.g. Sodemann and Zubler, 2010). But it is not straightforward to assess the relevance of different dynamical processes involved in moisture transport pathways solely based on trajectories. This is why a more sophisticated modelling approach is employed here.

---

## Author Comment (AC2) · 15 Mar 2019

**Contrasting stable water isotope signals from convective and large-scale precipitation phases of a heavy precipitation event in Southern Italy during HyMeX IOP13**

**By K. O. Lee et al.**

Reply to the referees' comments

In the following, the comments made by the referees appear in black, while our replies are in red, and the proposed modified text in the typescript is in blue.

Referee #2 comments

**General Comments**

A case study of a heavy precipitation event over southern Italy during HyMeX IOP13 in October 2012 has been discussed by far-reaching and detailed interpretation of model output. However the only validation of the results presented are numerous rain-gauge data from stations in southern Italy. Although the analysis of the model results are sophisticated and as far as possible reliable and in accordance with known synoptic and sub-synoptic flow pattern, any link to real processes is missing due to missing observations.

By the title the authors guide the reader to a case study of combined convective and large scale precipitation formation. But this is only partly the case. Studying the manuscript we can learn, that with the use of the selected models, how much in detail we can interpret and consequently use the COSMOiso model for small scale precipitation analysis and thus for forecasting issues. This should clearly be indicated in the title and it should be stated in the abstract that trajectory calculation and simulated stable water isotope analysis give valuable additional information for weather interpretation and forecasting purposes. The authors are asked to rephrase the title accordingly and to rewrite the abstract and conclusions. Otherwise, a case study based on model data only and without any supporting data (with the exception of the precipitation data) is not sufficient for publication.

Agreed. This study is entirely based on model simulations. We have been looking at SWI measurement obtained from space-borne retrievals, the Infrared Atmospheric Sounding Interferometer (IASI). IASI measurements provide continuous datasets in space at the global scale with coarse vertical resolution and limited precision. A recent retrieval algorithm (Lacour et al., 2017) provides δD in the middle troposphere (3-6 km column) with an error of 38 ‰ in cloud free regions. Figure A shows the IASI-retrieved δD and $q$ maps of the western Mediterranean on 15 October 2012. The enriched air mass (δD ≥ −150 ‰, $q$ ≥ 7 g kg⁻¹) is captured by IASI in the strait of Sicily (around 12–22°E, 30–38°N), as similarly as seen in COSMOiso simulation (see Figure 4 of manuscript).

[Figure]

**Figure A**. Maps of δD (left) and $q$ (right) measured by IASI on 15 October 2012 (both morning and evening orbit).

When comparing the IASI-retrieved δD and COSMOiso-derived δD in the western Mediterranean region (-10–25°E, 25–45°N) we see the IASI retrievals are biased high by more than 30 ‰ (Figure B). This comparison shows a reasonable agreement between IASI and COSMOiso, in spite of the bias. However, due to the temporally and spatially limited sample of IASI products associated with a single convection event, we have decided to use purely a modelling approach to demonstrate the usefulness of SWI data for understanding moist processes associated with a HPE. Correspondingly, the title, abstract, introduction, and conclusion have been corrected to specify this. Nevertheless, the comparison between IASI and COSMOiso provides a motivation for future studies to demonstrate the long-term (monthly to seasonal) SWI characteristics in connection to the HPE in the western Mediterranean basin.

[Figure]

**Figure B**. The IASI-retrieved δD and COSMOiso-produced δD in the western Mediterranean region on 15 October 2012 (both morning and evening orbit).

Accordingly, the following changes have been made:

♣ Page 1, line 2-3 (title)
"Contrasting stable water isotope signals from convective and large-scale precipitation phases of a heavy precipitation event in Southern Italy during HyMeX IOP 13: a modelling perspective"

♣ Page 1, line 15-17 (abstract)
"The dynamical context and moisture transport pathways embedded in large scale flow and associated with a heavy precipitation event (HPE) in Southern Italy (SI) are investigated with the help of stable water isotopes (SWIs) based on a purely numerical framework."

♣ Page 4, from line 23 (introduction)
"SWI measurements are mainly obtained from space-borne retrievals (e.g. Schneider et al., 2016; Lacour et al., 2017) and ground-based in-situ laser spectroscopy (e.g. Aemisegger et al., 2012). The space-borne measurements provide continuous datasets in space at the global scale with coarse vertical resolution and limited precision. On the other hand, ground-based measurements with high temporal resolution are only available from a few locations and from dedicated field campaigns. In particular, the data availability for the Mediterranean region is very limited. A notable exception is the airborne dataset acquired around Corsica (Sodemann et al., 2017) during the first Special Observing Period of the Hydrological cycle in the Mediterranean

Experiment (HyMeX SOP-1, Ducrocq et al., 2014). However, it does not include SWI observations for the days under scrutiny in this paper. Due to these limitations, we use a model to demonstrate the usefulness of SWI data for understanding moist processes associated with a Mediterranean HPE, for the first time."

♣ Page 16, from line 5 (conclusion)
"[…] Although our study is entirely based on a model simulation, the results suggest that the information on mesoscale moist dynamical processes and moisture transport that is contained in SWI, when combined with SWI observations, can provide very useful constrains on the representation of such processes in numerical models.

Our study is the first study to investigate the potential benefit of SWI in the context of a HPE in the Mediterranean. As such, our study provides a proof of concept of the usefulness of SWI data to understand the variety of origins and moist processes associated with air masses feeding the convection over SI. This will be further investigated in future research using SWI measurements obtained from various platforms, e.g. ground-based, near surface, airborne (Sodemann et al., 2017), and space-borne. Our modelling study will also allow designing forthcoming tailored field campaigns in the Mediterranean region."

After redirection the scope of the paper, precipitation process information based on trajectory and SWI model output can be a helpful tool to examine actual weather situations governed by mutual evaporation-condensation processes. This is relevant scientific research and within the scope of ACP as the paper introduces additional parameters in operational weather analyses. The relevant SWI literature is cited by the authors, and the model set up is explained sufficiently. The analysis of the model results is accurate and detailed. The description of the flow field and the evaporation-condensation cycle are reliable. The structure of the paper is comprehensible. Some comments on figures, abbreviations, text etc. see "minor points".
I recommend the paper for publication in ACP after redirecting the purpose of the paper.
We appreciate the time and effort you put in this review as well your mindful comments on our paper. We have worked hard to comply with all of them. Replies to each comment are listed below.

**Minor comment**
1. P1, L15 (new): Moisture transport pathways embedded in large scale flow and associated…
Corrected.

2. P2, L26: The paper "Diagnostic study of a HyMeX heavy precipitation event over Spain by investigation of moisture trajectories (Rohner, L., Nerding, K.-U., Corsmeier, U., 2016, Quarterly journal of the Royal Meteorological Society)" shows the analysis of a HPE by combination of simulations and observations including moisture trajectories. In "Sodemann et al. (2017)" the potential of airborne measurements of SWI (during HyMeX) is shown.
Thanks for noting the reference of Röhner et al., 2016. We have included the paper in the reference list (P2, L27), while Sodemann et al. (2017) and the potential of airborne measurement of SWI have been emphasized in P16, L14.

3. P3, L15: better: … condensation cycles during…
Corrected.

4. P3, L20: explain the expression in brackets in a separate sentence for better readability.
Corrected.

♣ From Page 3, line 19

"[…] For instance, low $\delta^2$H (typical range between −160 and −180 ‰) or $\delta^{18}$O (i.e. range between −20 and −30 ‰) values in atmospheric water vapour at surface indicate low air mass temperatures and strong rainout of air parcels (e.g. Jacob and Sonntag, 1991; Yoshimura et al., 2010), whereas high $\delta^2$H (typical range between −120 and −100 ‰) or $\delta^{18}$O (range between −18 and −14 ‰) indicate high air mass temperatures and recent admixture of fresh ocean evaporate. The $\delta$ notation describes the concentrations of the heavy isotopes relative to the isotope ratio of the Vienna Standard Mean Ocean Water– RVSMOW, by for instance, $\delta^{18}$O = (Rs/RVSMOW − 1) × 1000, where Rs = [H$_2^{18}$O]/[H$_2^{16}$O] is the isotope ratio of a water sample."

5. P4, third paragraph: abbreviations SI, IOP, P1, and P2 have been explained before in the abstract.
The abstract and the main text should be considered as 2 separate entities. Hence, acronyms need to be defined in the main text, even if they already appear in the abstract.

6. P6, L21: refer to Fig. 1
Corrected.

7. P6, L26: refer to Fig. 2
Corrected.

8. P8, L13: … transported from northern Africa…
Corrected.

9. P8, L15: isn't it 314−330 K instead of 308−326 K?
Corrected to 315−330 K with improved color scale of Fig. 3.

10. P8, L23: isn't it western edge of the Ty-box instead of north-eastern edge?
Corrected to western edge.

11. P8, last paragraph: this paragraph should be removed to section 4
Thanks for suggestions. The paragraph has been moved to the beginning of section 4.

12. P9, L9: The cold front is often mentioned in the text but never indicated in the figures (with exception of Fig. 14, conceptual model).
Agreed. The cold front is indicated where potential temperature ($\vartheta$) values show a large gradient (315–330 K) at 850 hPa and it is marked by a dashed line in right panels of Fig. 3.

[Figure]

**Figure 3.** Horizontal distributions of sea level pressure (shading) and geopotential height at 500 hPa (contours) (left), and potential temperature, $\vartheta$ (shading), and wind (black and white arrows) at 850 hPa (right) at 16 UTC (top) and at 20 UTC (middle) 15 October 2012, and 00 UTC on 16 October 2012 (bottom) produced by the COSMOiso simulation. Coastal line is depicted by black line. The location of cold front is depicted by a dashed line in right panels.

13. Section 4.1: The caption "Distribution of SWIs… prior to HPE" does not fit to the text: the interval with precipitation (16 UTC to 07 ITC) is discussed in this paragraph.

Agreed. We have restructured Sections 3 and 4. Section 4.1 has been combined with section 3, and the corresponding new subsection is entitled "Distribution of SWIs over the Mediterranean".

14. P12, L15: Figs. 8c and 8d do not exist.

Corrected to Figs. 10c, d.

15. P14, L24: … (grey encapsulated area in Fig. 14a). In the figure the area is called "blue encapsulated".

Corrected consistently to "green encapsulated area".

16. Figure 3: use isobars for showing the surface pressure field. Indicate the position of the cold front.

The sea surface isobar and geopotential height at 500 hPa have been shown in the left panels of Figure 3 to show the upper level trough. In the right panels, the horizontal wind at 850 hPa is indicated together with $\vartheta$ at 850 hPa, while the position of cold front is defined by a large $\vartheta$ gradient of 315–330 K (please see our reply for comment #12).

17. Figure 5: skip the upper and middle color bars.

As suggested the color scale has been adjust for sake of readability. With a suggestion from other referee, sections 3 and 4 have undergone restructuring to describe the core results more concisely. By this, Figure 5 have been re-ordered to Figure 4.

[Figure]

**Figure 4.** Horizontal distributions of water vapour mixing ratio at 850 hPa (left), $\delta^{18}O_v$ at 850 hPa (middle) and $\delta^{18}O_v$ at 600 hPa (right) at 16 UTC (top) and 20 UTC (middle) on 15 October 2012, and 00 UTC on 16 October 2012 (bottom).

18. Figure 8: a color bar is missing.
The color bar is now added to Figure 8.

[Figure]

**Figure 8.** Scatter diagram of $q$ and $\delta^{18}O_v$ along the backward trajectories of Figure 7 during (a) the times between −72 and −48 h, and (b) times between −48 and 0 h every 12 hours from 20 UTC on 15 October 2012. The colour of dot changes every 12 h. The mixing and Rayleigh lines are indicated in each panel by dashed and solid line, respectively. The averaged $q$ and $\delta^{18}O_v$ every 12 hours is displayed in the bottom right corner of each panel.

19. Figure 10: changing color bars for the sub-figures makes it difficult to interpret figure differences.
As suggested, a single color bar has been used for plots related to $\delta^{18}O_v$ (Fig. 10b, d, and f), and a single color bar was used $\delta^{18}O_r$ (Fig. 10c, e) and $\delta^{18}O_s$ (Fig. 10g).

[Figure]

**Figure 10.** Horizontal distributions of (a) surface hourly precipitation (mm), $\delta^{18}O_v$ (‰) at (b) model level 8 (about 542 m ASL), (c) model level 16 (about 2455 m ASL), and (d) model level 23 (about 5565 m ASL, $\delta^{18}O_r$ (‰) at (e) 542 m ASL and (f) 2455 m ASL, and $\delta^{18}O_s$ (‰) at 5565 m ASL at 20 UTC on 15 October 2012. Note that, due to the terrain-following coordinates, the SWI values are partly depleted over topography, e. g. in central Italy. The precipitating area is marked by the area enclosed by the dashed line.

20. Figure 13: as Fig. 10

Figure 13 has been improved in the same way of Figure 10.

[Figure]

**Figure 13.** Same as Figure 10 but for 00 UTC on 16 October 2012.

21. Figure 14: the color differences (grey, blue, yellow shading) is difficult to read if manuscript is printed.
Corrected to green, yellow, and red shading for better readability.

**Figure 14.** Schematics summarizing the main processes and the key features of water vapour isotopologues associated with deep convection upstream of SI and leading to the Phase 1 (a) and Phase 2 (b) of the HPE. In (a) and (b), white descending arrow indicate the mistral wind behind the edge of the cold front (thick black line). The white arrow in the yellow-shading encapsulated area illustrates the frontal wind at 850 hPa, and the white arrow in the red-shading encapsulated area (warm sea surface) indicates the elevated African moist plume. In (a), convective ascent and precipitating downdraft are depicted by red and blue arrows, respectively. In (b), the southern edge of upper trough is indicated by black dashed line and the cyclonic flow of the trough is indicated by grey dashed line.

---

## Author Comment (AC4) · 15 Mar 2019

**Contrasting stable water isotope signals from convective and large-scale precipitation phases of a heavy precipitation event in Southern Italy during HyMeX IOP13**

**By K. O. Lee et al.**

Reply to the referees' comments

In the following, the comments made by the referees appear in black, while our replies are in red, and the proposed modified text in the typescript is in blue.

Referee #3 comments

**General Comments**

The reviewed article is a thorough case study performed using COSMOiso. While the analysis of the simulation results is highly detailed, the title and abstract of the manuscript lead the reader to expect a comparison to observations. However, the only observations used are precipitation measurements. Considering the emphasis on isotope modelling, I would have expected at least some measurements of this nature. For that reason, the purpose of the manuscript as a whole, and especially the long and detailed analysis of isotope distributions within the manuscript, is not clear.

The main question is: what additional insight is gained from the use of a model-only isotope analysis. This problem is further underlined by the presented trajectory analysis, which seems to provide the same information which is derived from the isotope concentrations. For a recommendation for publication, the authors need to better outline the purpose of the study, as well as their reasoning for using COSMOiso and its advantages compared to a trajectory analysis or even just passive tracers.

Assuming these problems can be addressed, a number of other issues remain. Below is a detailed listing of major and minor comments which should help the authors to improve their manuscript substantially.

We appreciate the time and effort you put in this review as well your mindful comments on our paper. We have worked hard to comply with all of them. Replies to each comment are listed below.

**Major comment**

1. Section 4 is the core of the manuscript. However, even considering that, it seems out of proportion. It is not only very long but also difficult to read and descriptive over long spans. This sections would greatly benefit from being shorter and more concise.

Agreed. Sections 3 and 4 have undergone restructuring to describe the core results more concisely. The previous section 4.1 has been combined with section 3 to avoid the repetition about the synoptic context. Section 3 is now entitled "Overview of meteorological condition" with two subsections: "3.1 One HPE with two precipitation phases over southern Italy", and "3.2 Distribution of SWI over the Mediterranean". Sections 4 is now entitled "SWI distribution during two precipitation phases" with two subsections of "4.1 The convective phase of precipitation", and "4.2 The large-scale phase of precipitation".

2. Parts of the conclusion repeat contents of section 4 in too much detail, shorten and be much more concise and clear.

Agreed. The conclusion is now more concise after removing the content redundant with section 4.

♣ Page 15, from line 8 (conclusion)

"The 3-day backward trajectory analysis shows that the air parcels arriving in SI during P1 originate from the North Atlantic and descend within the upper-level trough over the north-western Mediterranean Sea. The SWI-depleted air mass (median $\delta^{18}O \leq -45$ ‰) within the descending air parcels  rapidly take up a large amount of water vapour from ocean evaporation (green encapsulated area in Fig. 14a) over the Tyrrhenian Sea and also from evaporated moisture from falling precipitation . Additional moisture is taken up over the Strait of Sicily  from mixing with the enriched moisture plume coming from Africa ($\delta^{18}O_v \geq -16$ ‰). The SWI-enriched low-level air masses arriving upstream of SI are convectively pumped to higher altitudes, producing precipitation over SI, and the SWI-depleted moisture is transported towards the surface within the downdrafts ahead of the cold front (red and blue arrows, Fig. 14a).

During P2 (Fig. 14b), just a few hours after P1, the origin of the air parcels arriving at SI is distinct, i.e. mostly from North Africa. The air parcels are moist and associated with large $\delta^{18}O$ values (bottom most arrow, median $\delta^{18}O_v \geq -16$ ‰). With the arrival of the upper-level trough ($\delta^{18}O_v \leq -45$ ‰ ) and  mistral ($\delta^{18}O_v \leq -25$ ‰ ) over the southern Tyrrhenian Sea, the strong cyclonic flow around the trough (grey dashed line in Fig. 14b) induces the advection of the moist plume towards SI and leads to large-scale uplift of the warm and moist African air mass along the cold front. It brings moisture and leads to gradual rain out of the air parcels over Italy . For the convective precipitation phase (P1), most of the moisture processes producing the HPE take place during the last 18 hours before the arrival over SI, while the large-scale advection of SWI-enriched moisture from the African plume by strong cyclonic flow lasts about 72 hours during the large-scale precipitation phase (P2). "

3. The authors need to discuss their results in a critical way which includes an explanation of the insights gained by using COSMOiso over a normal mesoscale simulation, which would be possible at a much higher resolution too. This discussion can be part of the last section Conclusions and Discussion.

The chosen setup (COSMOiso simulation with 7 km grid spacing and parameterized convection) is a tradeoff between high enough resolution for including detailed dynamics of the mesoscale systems and still being able to run efficiently over a large domain that also includes the moisture plume over North Africa. Also this selected resolution and large model domain reduce the dependence on the much coarser isotope boundary data (spectral resolution of T62 in IsoGSM) and enables us to calculate backward trajectories consistently over longer periods. This discussion is added in section 5, Conclusion.

♣ Page 16, from line 14

"[…] In this study, COSMOiso simulation at a horizontal grid spacing of about 7 km with parameterized convection results from a trade-off between having high enough resolution for including detailed dynamics of the mesoscale systems and being able to run efficiently over a large domain (about 4300 km × 3500 km) that includes the moisture plume over North Africa. This setup allows addressing the question we are interested in, namely: which isotope signals are due to local processes, and which are due to large-scale advection? To further study the details of the fractionation processes in and around deep convective systems, complementary investigations will be conducted using higher resolution convection-permitting simulation with a 2 km grid to shed a light on cloud microphysical processes inside deep convection."

4. Multiple figures are difficult to read, be it due to bad coloring or their size. The authors could greatly improve the manuscript's readability by making sure that figures use more contrasting color table, fewer contour levels and that the figure size and shape make better use of the available space. I mention specifics for certain figures throughout the minor comments, but the other figures can also be improved following those same guidelines.
Agreed. We have improved Figures 1, 3, 5, 8, 10, 12, 13, and 14 for better readability.

5. Figures are often referenced out of order, try to keep this to an absolute minimum. This will likely require some restructuring of the text.
The order of reference of Figures is now corrected throughout the paper.

**Minor comment**

1. P3, L1: large amounts of water vapor. How large?
Corrected.

♣ Page 3, from line 1
"[…] the intrusion of large amounts of moisture, about one quarter of the total integrated water vapour, […]"

2. P3, L10: observations of the most stable water isotopes alone can be limited this indicates that "normal" observations only look at this isotopes, but I assume that the sentence refers to classical observations which simply look at the total moisture without any regard for different isotopes. This should be more clear.
Corrected to "the SWI observation of other, less abundant SWIs, i.e. $H_2^{18}O$ and $HD^{16}O$".

3. P3, L14-15: replace *in the other phase (vapor)* with just *in vapor*
Corrected.

4. P3, L19-24: Please specify what *high* and *low* are in this context by giving typical values
The typical ranges of each values are indicated by referring to Jacob and Sonntag (1991) and Yoshimura et al. (2010).

♣ Page 3, line 19-23
"[…] For instance, low $\delta^2H$ values (typically ranging between −160 and −180 ‰) or low $\delta^{18}O$ values (i.e. ranging between −20 and −30 ‰) at the surface indicate air masses characterized by low temperatures and strong rainout of air parcels (e.g. Jacob and Sonntag, 1991; Yoshimura et al., 2010), whereas high $\delta^2H$ values (typically ranging between −120 and −100 ‰) or high $\delta^{18}O$ values (ranging between −18 and −14 ‰) indicate air masses characterized by high temperatures and recent admixture of fresh ocean evaporate."

5. P3, L29: remove commas
Removed.

6. P4, L16: change to *used **a** stable isotopic* or *used stable isotopic signal**s***
Corrected to "used stable isotopic signals".

7. P4, L20: add comma after mesoscale
Added.

8. P4, L25-27: move the part *that occurred (…) Mediterranean Experiment (HyMeX …)* to a separate sentence
Corrected while the acronyms of HyMeX has been defined in the previous paragraph.

♣ Page 5, line 7-10
"[…] The target HPE occurred during the Intensive Observation Period 13 (IOP 13) of the HyMeX SOP-1. Using a combination of ground-based, airborne and space-borne observations and numerical simulations of this HPE, Lee et al. (2016) investigated the detailed dynamic and thermodynamic environments of the two precipitation phases of the HPE."

9. P5, L1-5: This description is difficult to follow if one is not familiar with Lee et al. (2016). This should be moved to section 3, where it can make use of Figs. 1-3 for a thorough but concise description of the event (see also comments on Fig. 1)
Agreed. The description about Lee et al. (2016) has been moved to section 3.2.

♣ Page 9, line 2-7
"The moisture structure upstream of the HPE; 1) the presence of an African moisture plume favouring the efficiency of the convection to produce more precipitation, 2) the importance of southerly flow from the warmer Mediterranean Sea south of Sicily in enhancing the convergence ahead of the cold front, and 3) the role of the upper-level trough over southern France extending to the western Mediterranean in organizing convection at the leading edge of the surface front, highlighted by Lee et al. (2016) has been further studied using SWI data […]"

10. P5, L3: remove *wind* after *mistral*
Corrected.

11. P5, L4: change *convection activity* to *convective activity*
Corrected.

12. P5, L8: add *and* after the comma
Added.

13. P5, L10: change to *However, the origin and transport pathways of moisture have not been studied to date*
Corrected.

14. P6, L2: specify that this is a *deep* convection scheme, since the resolution of the model has not yet been mentioned at this point
Specified.

15. P6, L3: please add a very brief and concise description of what these physics and isotope parametrizations do, one to three sentences should suffice.
A brief description of isotope physics and parametrizations has been included.

♣ Page 6, line 7-18

"[…] All prognostic moisture fields, which are simulated by the model in terms of specific humidities, are duplicated twice, representing the specific humidities of $H_2^{18}O$ and $HD^{16}O$, respectively. From the prognostic specific humidity fields, the isotope ratios in usual δ-notation can be calculated. The heavy isotopes experience the same processes as the light isotope ($H_2^{16}O$), except during phase transition, when isotopic fractionation occurs. A one-moment microphysics scheme is used and deep convection is parameterised following Tiedtke (1989). In the microphysical scheme, transfer rates between the different water species during the formation of clouds and precipitation are specified. The heavy isotopes are affected by equilibrium fractionation during the formation of liquid clouds, and both non-equilibrium and equilibrium fractionation during the formation of ice clouds (using the predicted super-saturation) as well as the re-evaporation of rain drops. For the parameterisation of moist convection, all physical processes during simulated convective up- and downdrafts affect the heavy isotopes in a similar way as the standard light humidity, again taking into account equilibrium and non-equilibrium fractionation when appropriate."

16. P6, L11: please add some details about this model. Does it run operationally? Is it an analysis? Does it run only for specific cases?
The IsoGSM global simulation data is constrained to reanalysis data with the help of a nudging technique. The Scripps Experimental Climate Prediction Center's GSM was based on the medium range forecast model used at NCEP for making operational analysis and predictions. Isotope ratios in water vapour with a spectral resolution of T62 and on 17 vertical levels are obtained from the IsoGSM simulation. This information has been included in manuscript.

♣ Page 6, from line 25

"[…] For the water isotopes, initial and boundary data are taken from a historical isotope global circulation model IsoGSM (which is based on the Scripps Experimental Climate Prediction Center's GSM that was used operationally for medium range forecasts at NCEP) simulation by Yoshimura et al. (2008), who performed these simulations using a nudging technique (see also Pfahl et al., 2012). The Scripps Experimental Climate Prediction Center's GSM was based on the medium range forecast model used at NCEP for making operational analysis and predictions."

17. P6, L12-16: Why is a resolution of 7 km used? Is this to be able to differentiate between convective and other precipitation by using the convection scheme's precipitation? Resources? Other reasons? Please specify.
This resolution was used for operational predictions at the German Weather Service DWD for a long time, such that the model is very well tuned in this configuration. We also chose this relatively coarse resolution because it allows for a large model domain that reduces the dependence on the much coarser isotope boundary data (spectral resolution of T62 in IsoGSM) and enables us to calculate backward trajectories consistently over longer periods.

♣ Page 16, from line 14

"[…] In this study, COSMOiso simulation at a horizontal grid spacing of about 7 km with parameterized convection results from a trade-off between having high enough resolution for including detailed dynamics of the mesoscale systems and being able to run efficiently over a large domain (about 4300 km × 3500 km) that includes the moisture plume over North Africa. This setup allows addressing the question we are interested in, namely: which isotope signals are due to local processes, and which are due to large-scale advection? To further study the details of the fractionation processes in and around deep convective systems, complementary investigations will be conducted using higher resolution convection-permitting simulation with a 2 km grid to shed a light on cloud microphysical processes inside deep convection."

18. P6, L22: 5 days trajectories in a 5 day simulation? So just back to the start of the simulation or until they leave the domain?

The trajectories are computed back in time until they leave the domain. The sentence has been corrected for sake of the readability.

♣ Page 7, line 12-13

"[…] The trajectories are computed five days back in time. Note that generally the COSMO trajectories move out of the regional model domain after 3 days."

19. P7, L6-8: Why are these values chosen?

The values are chosen based on the near sea surface temperature of SI region.

20. P7, L20: Spell out *two* in the section title

Corrected.

21. P7, L23: hour is abbreviated with just *h*

Corrected also at other places.

22. Fig.1: Some dots have edges, others have none. Remove all edges and make sure the higher precipitation measurements are plotted on top of the lower values to keep them clearly visible and not hide maxima. The Figure is very small and the comparison is difficult to read. In Fig. 1b all contour colors from 5 to 25 mm look almost exactly the same in print, please use a color table which shows the differences more clearly. Fewer levels might help to achieve better contrast, do you really need 24 different ones?

As suggested the edges have been removed and the size of color dots has been enlarged, and the color scale has been adjusted for sake of readability.

♣ Page 23

[Figure]

**Figure 1.** Accumulated precipitation during IOP 13 from 00 UTC on 15 October 2012 to 03 UTC on 16 October 2012 obtained from (a) rain gauge network, and (b) COSMOiso simulation.

23. P7, L26: *a large precipitation* rephrase
The sentence has been rephrased.

♣ Page 8, line 17
"shows precipitation in excess of 10 mm […]"

24. P7, L24-27: This sentence is too long and convoluted, simplify by moving the total precipitation amount to a separate sentence.
This sentence has been divided into two sentences.

♣ Page 8, line 15-20
"[…] The temporal evolution of the COSMOiso domain-averaged total precipitation within the SI area (bars in Figure 2) shows precipitation in excess of 10 mm within SI between 19 UTC on 15 October and 01 UTC on 16 October. The period has two distinct precipitation phases: 1) a convective precipitation phase (**P1**) in the late afternoon (19–21 UTC) on 15 October (dashed line in Fig. 2), and 2) a large-scale precipitation phase (**P2**) just before midnight (22–00 UTC) on that day (solid line). […]"

25. P8, L8: remove comma after *France*
Corrected.

26. Fig. 3: Change the colored contours of MSLP to lines, chose a good interval (not too dense) and smooth the field a bit if necessary. Add colored contours of 500 hPa geopotential to show the position of the trough. Move the vectors from the left panels to the right panes, they contain information of the same level.
The 500 hPa geopotential height is contoured on the shaded area of MSLP in the left panels of Figure 3.

[Figure]

**Figure 3.** Horizontal distributions of sea level pressure (shades) and geopotential height at 500 hPa (contour) (left), and potential temperature, $\vartheta$ (shades), and wind (black and white arrows) at 850 hPa (right) at 16 UTC (top) and at 20 UTC (middle) 15 October 2012, and 00 UTC on 16 October 2012 (bottom) produced by the COSMOiso simulation. Coastal line is depicted by black line. The location of cold front is depicted by a dashed line in right panels.

27. P8, L11, 12: don't use *very* in scientific text, be specific.
Corrected.

♣ Page 9, line 10-15
"[…] winds associated with cold and dry air, with $\delta^{18}O_v$ less than −16‰ and $q$ less than 2 g kg$^{-1}$ (Fig. 4a, b), and thus low potential temperature, $\vartheta$, are located over the Gulf of Lion (≤ 302 K, dark-blue area in Fig. 3b). […] the African moist plume with values of $\vartheta \geq 330$ K […]"

28. P8, L13: change *high $\vartheta$ values (≥ 330 K)* to *values of $\vartheta \geq 330$ K*
Corrected.

29. P8, L16, 17: Could convection be causing the cool areas in the 850 hPa potential temperature map over TY?
The original sentence was "Over the Tyrrhenian Sea ('TY' box in Fig. 1b), upstream of SI, a large horizontal $\vartheta$ gradient (315–330 K) can be seen at 850 hPa, indicating the elongation of the surface cold front along a southwest to northeast axis".
The convection positioned at the southern edge of this front, where high values of $\vartheta$ (≥ 325 K) are seen.

30. P8, L17: It is never explained that the model does, in fact, not produce two peaks. They are only visible when separating precipitation from the convection scheme and precipitation produced by microphysics.
The model, in contrast to the observations, does not produce two peaks in the total precipitation. These peaks can be seen by looking at the two precipitation types separately. This criticism has been included in manuscript.

♣ Page 8, line 24-26
"[…] P1 is related to rain from the convection parameterization, and P2 is related to rain associated with large-scale vertical motion. The model, in contrast to the observation, does not produce two peaks in the total precipitation. These peaks can be seen by looking at the two precipitation types separately."

31. P8, L18: the trough is never shown, add reference to Fig. 3 after adding the 500 hPa geopotential as suggested.
In the left panels of Figure 3, the 500 hPa geopotential height is contoured on the shaded area of MSLP. Please see the answer for comment #26.

32. P8, L23: *strong cyclonic flow* there is only one arrow within the box, curvature is hard to see.
The Figure 3 (right panels) has been corrected to see better the cyclonic flow, but very weak wind at the core region is not displayed for sake of readability.

33. P9, L12, 13: *very low* and *large* are not helpful in this context, just use the values. However, a short explanation on why the threshold between these two values is important would be helpful.
Corrected.

♣ Page 9, lines 10 and 16
"[…] cold and dry air, with $\delta^{18}O_v$ values less than −16 ‰ […]$\delta^{18}O_v$ values in excess of −25 ‰ can be seen […]"

34. P9, L26: the front is not really close to SI
The cold front is indicated where potential temperature ($\vartheta$) values show a large gradient (315–330 K) at 850 hPa and it is marked by a dashed line in right panels of Fig. 3. We can see the front is close to SI.

35. P10, L27: *mostly very dry* use values instead, be specific
Corrected.

"[…] These air parcels are mostly dry ($q \leq 5$ g kg$^{-1}$) along the track during the 3 days […]"

36. P11, L2: remove *the* before q and $\delta^{18}O_v$, remove *values* after $\delta^{18}O_v$
Corrected.

37. P11, L6: change to *The median q value (…) factor of 2.5*
Corrected.

38. Fig 7: Figure has lots of white space and the way the map is shown causes even more. Try to reduce this to make the important parts a bit larger.
We have reduced much of white space and enlarged the figure.

[Figure]

**Figure 7.** History of air parcel arriving at **SI** in layer of 800–700 hPa at 20 UTC on 15 October 2012. (a) water vapour mixing ratio, $q$ (g kg$^{-1}$), (b) $\delta^{18}O_v$ (‰), (c) surface evaporation (mm h$^{-1}$), (d) altitude (km), and (e) time (h).

39. P11, L12: replace *the average q* with *q is about* 9 g kg$^{-1}$ *on average*
Corrected.

40. P11, L14: It is never explained what a Rayleigh line is
Corrected.

41. P11, L19-21: This sentence is complicated. Also why?
The sentence was "This shows that the descending air parcels mix with the air parcels from lower altitudes, and near surface air parcels mix between surface evaporation and background vapour".
Figure 9a, b show that the lower to upper-level trajectories (0.1–7 km altitudes) follow a mixing line during their descent, and this indicates that the descending air parcels from upper levels mix with the air parcels from lower altitudes. The descent of drier air parcel to near the sea surface also increases evaporation. For sake of clarity, this sentence has been improved.

♣ Page 12, line 8-9
"[…] This shows that the descending dry air parcels mix with the warm and moist air parcels from lower altitudes, which also increases surface evaporation."

42. P11, L28: replace *many* with *multiple*, change *convection* to *convective*
Corrected.

43. P12, L2: usage of low/high seems inconsistent looking at the numbers
The sentence has been rewritten.

♣ Page 12, line 18-19
"[…] Within the precipitation area, relatively lower $\delta^{18}O_v$ values (≤ −16 ‰) than in the vicinity are found at 542 m ASL while relatively high $\delta^{18}O_v$ values between −20 and −24 ‰ are found at 2455 m and 5565 m ASL […]"

44. P12, L13: change to *with values of* $\delta^{18}O_v$ *larger than;* replace *toward* with *around*
Corrected.

45. P12, L18-19: rewrite sentence
Corrected.

♣ Page 13, line 7-8
"The Lagrangian analysis indicates that most of the processes inducing precipitation during P1 take place during the last 18 hours over the Tyrrhenian Sea and the Strait of Sicily."

46. P12, L25: change to *convective mixing injects SWI-enriched moisture into higher altitudes*
Corrected.

47. Fig. 8: explain colors in the caption, some dots have edges and others don't, figure is small, you could change the aspect ratio to fit the page width for better readability. This also applies to the other figures of this type (9 and 12).
As suggested the Figure 8 has been improved for better readability. However we kindly propose to keep the aspect ratio of the figure to better identify the characteristics.

[Figure]

**Figure 8.** Scatter diagram of $q$ and $\delta^{18}O_v$ along the backward trajectories of Figure 7 during (a) the times between −72 and −48 h, and (b) times between −48 and 0 h every 12 hours from 20 UTC on 15 October 2012. The colour of dot changes every 12 h. The mixing and Rayleigh lines are indicated in each panel by dashed and solid line, respectively. The averaged $q$ and $\delta^{18}O_v$ every 12 hours is displayed in the bottom right corner of each panel.

[Figure]

**Figure 9.** Scatter diagram of $q$ and $\delta^{18}O_v$ along the backward trajectories of Figure 7 but for all altitudes of 1−2 km (black dots), 2−3 km (red dots), 3−4 km (yellow dots), 4−5 km (green dots), 5−6 km (blue dots), and 6−7 km (purple

dots) at (a) −6 h, (b) −3 h, and (c) 0 h from 20 UTC on 15 October 2012. The mixing and Rayleigh lines are indicated in each panel by dashed and solid line, respectively.

[Figure]

**Figure 12.** Scatter diagram of $q$ and $\delta^{18}O_v$ along the backward trajectories of Figure 11 during (a) the times between −72 and −48 h, (b) times between −48 h and −24 h, and (c) times between −24 h and 0 h from 00 UTC on 16 October 2012 every 6 hours. The colour of dot changes every 6 h. The mixing and Rayleigh lines are indicated by dashed and solid line, respectively. The averaged $q$ and $\delta^{18}O_v$ every 6 hours is displayed in the bottom right corner of each panel.

48. Fig. 10: panel titles say *vapour* and *rain water*, change them to clearly indicate that they show $\delta^{18}O_v$ **for** vapour/rain water. Also, model levels are not at a constant height. Does this have any effect over mountains? If so, explain which one? Better alternative: plots for certain altitudes above sea level, e.g. 500, 2500, and 5000 m, instead of model levels.

The panel titles have been corrected to $\delta^{18}O_v$, $\delta^{18}O_r$, and $\delta^{18}O_s$ correspondingly. The figure 10 shows $\delta^{18}O_v$, $\delta^{18}O_r$, and $\delta^{18}O_s$ at a model level 8, 16, and 23 which altitudes are about 542 m, 2455 m, and 5565 m above the sea surface, respectively, in regions without topography. However, as noted by the reviewer, the model levels follow the terrain and the fields are thus shown for different altitudes over topography. As the precipitation in SI region occurred mostly near the coast, and the associated moisture processes occurred over the Tyrrhenian Sea and Strait of Sicily, we keep the plot as it is, but a note has been added to the caption.

[Figure]

**Figure 10.** Horizontal distributions of (a) surface hourly precipitation (mm), $\delta^{18}O_v$ (‰) at (b) model level 8 (about 542 m ASL), (c) model level 16 (about 2455 m ASL), and (d) model level 23 (about 5565 m ASL, $\delta^{18}O_r$ (‰) at (e) 542 m ASL and (f) 2455 m ASL, and $\delta^{18}O_s$ (‰) at 5565 m ASL at 20 UTC on 15 October 2012. Note that, due to the terrain-following coordinates, the SWI values are partly depleted over topography, e.g. in central Italy. The precipitating area is marked by the area enclosed by the dashed line.

[Figure]

**Figure 13.** Same as Figure 10 but for 00 UTC on 16 October 2012.

49. P13, L2: replace *over* with *from*
Corrected.

50. P13, L3: Threshold of 5 g kg$^{-1}$ is not visible in the figure
The value is obtained from an average over all trajectories, not those shown in Figure 11. For better understanding, the sentence has been corrected.

♣ Page 13, line 20-21

"The air parcels are consistently moist along the tracks (Fig. 11a), with average $q$ value mostly ≥ 5 g kg⁻¹ along the track, in contrast […]"

51. P13, L7: replace *for instance* with *and*
Corrected.

52. P13, L7, 8: Use a non-breaking space between multiple units, in LaTeX to avoid line breaks between them. This can be done by using ~ instead of a pace like this: 9~g~kg or in MS word by using Ctrl + Shift + Space
We appreciate the guide. The non-breaking space has been used entire manuscript.

53. P13, L20-21: rephrase, also, do not use *precipitation cell* unless explicitly referring to a single convective cell
Corrected.

♣ Page 14, line 8-10

"At 00 UTC on 16 Oct. during P2, stronger precipitation than that of P1 is produced, and the precipitation system is located mainly over SI (marked area closed by dashed line in Fig. 13a). In the vicinity of the precipitating region, strong cyclonic south-westerly flow ≥ 25 m s⁻¹ is dominant at 2455 m and 5565 m ASL (Fig. 13d, f)."

54. P13, L24: The depletion is hardly visible at 5500 m
The depletion at 5565 m is visible with the improved color scale of Figure 13.

55. P14, L1-5: going back to earlier Figures is tedious and disrupting, try to avoid if possible by restructuring the text
Agreed. However we kindly propose to keep as it is to complete our comprehension from entire analysis.

56. P14, L6-8: rephrase sentence
Corrected.

♣ Page 14, line 24-25

"The Lagrangian analysis indicates that the moistures that feeds the convection during P2 is coming from North Africa and the air parcels take up additional moisture (2–3 g kg⁻¹) over the Mediterranean."

57. P14, L9: replace *entrainment* with *mixing,* entrainment is usually used in the context of convective updrafts
Corrected.

58. P14, L15: replace *convective* with *convection*
Corrected.

59. P14, L20: the three paragraphs starting here are especially long and too descriptive, be more concise. Do not simply repeat details from previous sections in the conclusions.
The three paragraphs become concise by removing the repeated description.

60. P15, L4, 5: do not use formulations like *totally different* in scientific texts
Corrected to "arriving at SI is *distinct*".

---

## Referee Report (RR1)

**Second review of**

*Contrasting stable water isotope signals from convective and large-scale precipitation phases of a heavy precipitation event in Southern Italy during HyMeX IOP 13*

February 2019

**Overview**

This review comments on the revised version of the article. The authors have made a great effort to address the comments and rectify the the substantial weaknesses of the manuscript, which were present in the first version. All major and minor points were adequately taken care of and I recommend publication of the manuscript after a number of minor revisions. Wile the comments below are numerous, none of them warrant another round of major revisions, as all of them can be addressed without investing too much time.

**Minor Comments**

1. While reading the manuscript, I still missed the basic justification for this article, as I have emphasized in my first round of comments. However, it turned out to be located in the last paragraph of the last section of the article. I suggest to present it toward the end of Section 1 and even make a short comment on the intentions behind this study in the abstract!

2. In section 2.1, the grid spacing and domain dimensions are mentioned at the very end of the section (p. 7, l. $2 - 6$). I suggest to move them to the start of the model description or at the very least to the beginning of the second paragraph.

3. While section 3 is greatly improved, the shortening had an unwanted side effect. Due to the density of the text, the third and fourth paragraph of section 3.2 (p. 10, l. $6 - 28$) contain no less than 12 references to figures, more than one every two lines, referring to four different figures in the order 5, 3, 5, 3, 3, 4, 3, 4, 4, 5, 5, and then back to 2. I understand that it is necessary to refer the reader to the correct figure, but you should try to rearrange these two paragraphs a bit, such that they don't jump around between figures this often and contain at least slightly fewer references to figures.

4. The manuscript contains multiple instances of *the X value*, where $X$ can be a quantity like $q$ or $\theta$ and so on, remove both, *the* and *value*, and just use $X$ instead (e.g. p 11, l. 4, 5, 6, 7, 27, 28–29, ...)

5. It says on p. 11, l. 20 that *between 18 and 6 hours before arrival...* but looking at Fig. 7 shows evaporation more or less the instant the trajectories start passing over the sea, for some of them well above 24 hours before the event. Perhaps you should consider a lest restrictive formulation here.

6. Section 4.1.1, third paragraph refers to dots following mixing/Rayleigh lines, but that doesn't really seem to be the case, at least not exactly. Do these lines depend on the constants in their equation, which might fit better for other values? If so, consider using them. If not, or if this is too much effort, explain more clearly how they agree with the scattered dots.

7. At multiple locations in the text, a moist plume over Africa is mentioned, and it is not always referred to in the same way (African moist plume, African moisture plume, the plume of ...), please keep this consistent throughout the manuscript. Also, explain whether this is a climatological plume or a a feature of this specific event.

8. Section 5 can still be slightly shortened by removing some of the values (not needed in the conclusions) and condensing the text further, specifically the part on p. 15, l. 8 – 26.

**Typos and Formulation**

| page | line(s) | comment |
| --- | --- | --- |
| 1 | 20 | replace *linked to a frontal feature* with *ahead of a cold front* |
| 2 | 4 | change to *large amount of convective precipitation* |
| 2 | 6 | replace *preceding the occurrence of* with *before* |
| 2 | 8 | remove comma after *SI* |
| 2 | 9 | replace *brings* with *lifts*, remove *masses* |
| 2 | 10 | remove *to higher altitudes* |
| 2 | 23 | change *mesoscale deep convection* to *organized deep convection*, add comma after *events*, remove *along* |
| 2 | 24 | remove *content*, remove *deep* |
| 2 | 25 | replace *question* with *research topic* |
| 2 | 28 – 29 | Using *either... or* like this indicates that Africa or extratropical remnants of Atlantic tropical cyclones over the Atlantic are the only sources of moisture, but other sources are possible as well as a combination of sources. I suggest to rephrase this a bit to account for these possibilities or at least not exclude them. |
| 2 | 29+ | Change sentence to *More recent studies (e.g. Lee et al., 2016 and 2017), pointed out a significant moisture contribution, one quarter of the total integrated water vapor, from North Africa...* |
| 3 | 9 | remove the *s* at the end of *isotopes* |
| 3 | 11 | replace *constraints* with *insights* |
| 3 | 20 | change *at surface* to *at the surface* |
| 4 | 11 | remove comma after *evaporation* |
| 4 | 15, 17 | repetition of *the free troposphere of ...* |
| 5 | 12 | change *in the low levels* to either *in the lower troposphere* or simply *at low levels* |

| page | line(s) | comment |
| --- | --- | --- |
| 5 | 12 | What exactly is meant by *initiated over Algeria*? Is this where convection first occurred? |
| 5 | 13 | replace *large* with *high* |
| 5 | 14 | add *s* after *temperature* |
| 6 | 5 | replace *sense* with *context* |
| 7 | 15 | replace *written as an output* to *extracted from the model* or just change to *written out* |
| 7 | 20 | insert *specific* before *humidity*, remove comma after *humidity* |
| 8 | 4 | add *s* after *ratio* |
| 8 | 10 | add *s* after *condition* |
| 8 | 17 | insert *region* after *SI* |
| 8 | 23 | replace *a reasonable* with *good* |
| 8 | 25 | add *s* after *observation* |
| 9 | 2 – 6 | The two parts of the sentence *The moisture structure (...) has been further studied...* are extremely separated and the sentence becomes difficult to read. You could rephrase it to something like: *The moisture structure upstream of the HPE studied by Lee et al. (2016) has been further analyzed. Three features are highlighted below: 1) the presence of ...* |
| 9 | 7 | add commas before and after *located over south-eastern France* |
| 9 | 8 | The threshold of 1002 seems arbitrary, this might be a relic from back when the area of MSLP $< 1002$ hPa was the only shaded part of Fig. 3. |
| 9 | 10 | remove comma after *air* |
| 9 | 11 | remove commas before and after *q*, remove *thus* |
| 9 | 15 – 16 | change to *(...) is located ahead of the trough (red area in Fig. 3b).* |
| 9 | 16 | change to *Comparing the maps of q and $\delta^{18}O_v$ (crescent enclosed by a dashed line in Fig. 4a–b) reveals an additional...* |
| 9 | 28, 29 | repetition of *the hourly evolution of* |
| 10 | 3 | move *slightly* to after *increase* |
| 10 | 7 | is it the upper level trough or the cold front? Those are two different things. |
| 10 | 12 | remove both, *the* and *level*, at the end of the sentence |
| 10 | 13 | remove *Then* at the beginning of the sentence, add a comma after *UTC* |
| 10 | 14 | which western edge? |
| 10 | 27 | insert *at* after *SI* |
| 11 | 2 | insert *the* before *two* |
| 11 | | Multiple instances of *history* where *evolution* might be the better word (e.g. lines 7 and 11) |
| 11 | 10 | change title to *Phase one: the convective phase* |
| 11 | 13 – 15 | change to *The 3-day backward trajectories in Fig. 7 indicate that the air parcels arriving at SI in the $800 – 700$ hPa layer originated over the North Atlantic.* |
| 11 | 16 | replace *are mostly* with *remain* |
| 11 | 17 | add comma after *SI*, remove *rapidly* |
| 11 | 20 | change *as well as* to *and* |
| 11 | 22 | change *mixing* to *they mix* |

| page | line(s) | comment |
|---|---|---|
| 11 | 23 | remove *occurs*, change *the median q value* to *the median of q* |
| 11 | 27 | change to *(...) before their arrival, showing that q and $\delta^{18}O_v$ increase rapidly in the last 12 hours before the parcels arrive over SI. Between 60 and 12 hours before their arrival (Fig. 8a, b), q and $\delta^{18}O_v$ are still relatively small, at around 2–6 g kg$^{-1}$ and between -25 and -19 ‰, respectively.* (Is this really in the dry pocket of upper-level trough? The trough itself should also be somewhere else this long before the event... |
| 12 | 3 | change to *for conditions in the Mediterranean* |
| 12 | 3 | *blue line* I do not see a blue line in the figure, does this refer to the dashed line? |
| 12 | 6 | *gray to purple* - shouldn't this be green to purple? |
| 12 | 14 | change to *Horizontal SWI distribution* |
| 12 | 15 | remove *feature*, remove *a region of enhanced convective activity and* |
| 12 | 16 – 18 | change *multiple convective cells* to *a convective line, which extends (...) and is located ahead of the surface cold front. Westerly and north-westerly winds prevail at 542 m ASL while south-westerly wind is dominant at 2455 m ASL.* |
| 12 | 18 – 21 | change to *Within the precipitation area, lower $\delta^{18}O_v$ values (< -16‰) than in the vicinity are found at 542 m ASL while locally higher $\delta^{18}O_v$ values are found at 2455 m and 5565 m ASL (Fig. 10d, f), indicating the presence of strong and deep convective mixing. This convection causes the vertical transport of SWI-depleted moisture ...* |
| 13 | 1 | remove *large* |
| 13 | 7 | remove *of* |
| 13 | 9 | replace *at* with *along* |
| 13 | 10 | change to *Additional moisture is then...* |
| 13 | 12 | add *s* after *hour*, remove *the ... values* |
| 13 | 13 | change *strongly increase* to *increase strongly* |
| 13 | 19 | remove third zero |
| 13 | 23 | add comma after *altitudes* |
| 13 | 24 | the vapor take-up isn't over North Algeria, it's over the Mediterranean sea |
| 13 | 24 | end sentence after *(Fig. 11a–b)* and remove *and* before *the median q*, change to *the median of q* |
| 13 | 25 – 26 | change *precipitation onset* to *the onset of precipitation* |
| 13 | 26, 27 | repetition of *moist and SWI-enriched* |
| 14 | 2 | add *s* at the end of *parcel*, add comma after *particular* |
| 14 | 7 | change to *Horizontal SWI-distribution* |
| 14 | 12 | this is still hardly visible at 5500 m, all shades of blue look almost exactly the same in print |
| 14 | 15 | the black line shows very little decrease, it's visible for the others though |
| 14 | 19 | multiple repetitions of *moisture plume over North Africa* or similar in this line |
| 14 | 23 | replace *becomes more depleted* with *decreases* |
| 14 | 24 | replace *that* with *which* |

| page | line(s) | comment |
| --- | --- | --- |
| 14 | 24 | I thought P2 was dominated by large scale ascent, not convection |
| 14 | 24 – 28 | This paragraph seems like it belongs to section 4.2.1, not 4.2.2 |

---

## Author Response (AR2)

**Contrasting stable water isotope signals from convective and large-scale precipitation phases of a heavy precipitation event in Southern Italy during HyMeX IOP13**

**By K. O. Lee et al.**

Reply to the referees' comments

In the following, the comments made by the referees appear in black, while our replies are in red, and the proposed modified text in the typescript is in blue.
* * *
| Referee comments |
| --- |

**Overview**

This review comments on the revised version of the article. The authors have made a great effort to address the comments and rectify the substantial weakness of the manuscript, which were present in the first version. All major and minor points were adequately taken care of and I recommend publication of the manuscript after a number of minor revision. While the comments below are numerous, none of them warrant another round of major revisions, as all of them can be addressed without investing too much time.

We appreciate the time and effort you put in this review as well your insightful comments to improve our paper. Replies to each comments are listed below.

**Minor Comments**

1. While reading the manuscript, I still missed the basic justification for this article, as I have emphasized in my first round of comments. However, it turned out to be located in the last paragraph of the last section of the article. I suggest to present it toward the end of Section 1 and even make a short comment on the intentions behind this study in the abstract!

As suggested, we present the basic justification of this article: i) earlier at the end of Section 1, and ii) shortly in the abstract.

♣ Page 5, lines 21–26 (end of Section 1)

"Here we investigate these moisture transport processes using trajectory calculations and SWI data obtained from a COSMOiso numerical simulation with 7-km horizontal resolution with parameterized convection. This setup results from a trade-off between having high enough resolution for including detailed dynamics of the mesoscale systems and being able to run efficiently over a large domain that includes the moisture transport from Africa. More importantly, it allows addressing the question we are interested in, namely: which isotope signals are due to local processes, and which are due to large-scale advection? […]"

♣ Page 1, lines 21–25 (Abstract)

"[…] The moisture transport and processes responsible for the HPE are analysed using a simulation with the isotope-enabled regional numerical model COSMO$_{iso}$. The simulation at a horizontal grid spacing of about 7 km over a large domain (about 4,300 km × 3,500 km) allows to distinguish the isotopes signal due to local processes or large-scale advection. […]"

2. In section 2.1, the grid spacing and domain dimensions are mentioned at the very end of the section (p. 7, l. 2–6). I suggest to move them to the start of the model description or at the very least to the beginning of the second paragraph.

Agreed. The grid spacing and domain dimension are now mentioned at the beginning of the second paragraph of Section 2.1.

♣ From Page 6, line 26
"In this study, a horizontal grid spacing of 0.0625° (in a rotated grid), corresponding to about 7 km is used with 40 hybrid vertical levels. The model domain covers the northwestern Mediterranean, the east Atlantic, and the northern African regions (longitude ranging from −16.3 to 22.8°E and latitude ranging from 17.3 to 49.2°N, i.e. about 4,300 km × 3,500 km). […]"

3. While section 3 is greatly improved, the shortening had an unwanted side effect. Due to the density of the text, the third and fourth paragraph of section 3.2 (p. 10, l. 6–28) contain no less than 12 references to figures, more than one every two lines, referring to four different figures in the order 5, 3, 5, 3, 3, 4, 3, 4, 4, 5, 5, and then back to 2. I understand that it is necessary to refer the reader to the correct figure, but you should try to rearrange these two paragraphs a bit, such that they don't jump around between figures this often and contain at least slightly fewer references to figures.

The paragraph has been re-arranged to avoid jumping to different figures.

♣ From Page 10, line 10
    " At 20 UTC (**Fig. 3**c, d), southerly winds (10−15 m s$^{-1}$) transport the warm and moist air mass with high $\vartheta$ values (≥ 325 K) from the Strait of Sicily to SI, and the convection occurred in the high $\vartheta$ region at the southern edge of the front (dashed line in **Fig. 3**d). The frontal wind convergence of south-westerly and southerly winds (10−15 m s$^{-1}$) can be seen upstream of the HPE at the 850-hPa level. Meanwhile, the African moisture plume including the SWI-enriched air mass ($q ≥ 10$ g kg$^{-1}$ and $\delta^{18}O_v ≥ −16‰$ in **Fig. 4**d–f) continues to advect toward SI.

    At 00 UTC when the trough is located in the southern Tyrrhenian Sea with the low-level mistral air mass ($q ≤ 3$ g kg$^{-1}$ and $\delta^{18}O_v ≤ −24‰$ in **Fig. 4**g–h) at the western edge, strong cyclonic flow can be identified over the SI region while the warm and moist air mass ($\vartheta ≥ 325$ K) over the Strait of Sicily is continuously advected towards SI (**Fig. 3**f). Higher up, at 600 hPa, the trough-related, strongly SWI-depleted air masses descending from higher altitudes show $\delta^{18}O_v$ values lower than −45 ‰ (**Fig. 4**i). In contrast to the trough, the African moisture plume is associated with large $q$ values in excess of 10 g kg$^{-1}$ at 850 hPa level extending to the SI region (**Fig. 4**g).

    During the two precipitation phases at 20 UTC and 00 UTC, both $\vartheta$ and $\delta^{18}O_v$ drop dramatically in the TY region with the arrival of the upper-level trough and cold front (**Fig. 5**a), while the warm and moist air mass with large $q$ and large $\delta^{18}O_v$ coming from tropical Africa persists upstream of SI (**Fig. 5**b). As $\vartheta$ decreases from 322 to 300 K in TY (**Fig. 5**a), the $\delta^{18}O_v$ drops more rapidly at altitudes above 3 km compared to the $\delta^{18}O_v$ drop seen in lower altitudes, where the trough-related dry airstreams are moistened by SWI-enriched fresh ocean evaporate. The minimum $\delta^{18}O_v$ value increases lowering the altitudes to near surface, for instance, the minimum $\delta^{18}O_v$ values of −23 and −36 ‰ are seen at 1−2 and 2−3 km ASL respectively, while values lower than −47 ‰ occur at altitudes above 3 km ASL. The hourly evolution of average $\delta^{18}O_v$ in the TY region shows the propagation of the surface front and upper-level trough at altitudes of 1−7 km ASL, and the associated subsidence of dry and cold air. It is worth noting that the arrival timing of cold and dry air subsidence in TY, 19−20 UTC, (**Fig. 5**a) corresponds to the onset of precipitation in SI, 19 UTC (vertical bars, **Fig. 2**)."

4. The manuscript contains multiple instances of the X values, where X can be a quantity like *q* or $\theta$ and so on, remove both, *the* and *value*, and just use X instead (e.g. p. 11, l. 4, 5, 6, 7, 27, 28–29, …).
Throughout the manuscript, *the* and *value* have been removed. For instance,

♣ Page 11, lines 8–11
"While  *q*  increases gradually to 13.5 g kg$^{-1}$ until 19 UTC, just before P1,  $\delta^{18}O_v$  maximizes to –13.6 ‰ at 16 UTC and then decreases during P1 to –15 ‰. During P2,  $\delta^{18}O_v$  increases shortly to –14.6 ‰ whereas  *q*  continues to decrease to 8 g kg$^{-1}$."

5. It says on p. 11, l. 20 that *between 18 and 6 hours before arrival*… but looking at Fig. 7 shows evaporation more or less the instant the trajectories start passing over the sea, for some of them well above 24 hours before the event. Perhaps you should consider a less restrictive formulation here.
Agreed. The evaporation occurs instantly passing over the sea surface during 18−6 hours or 24−18 hours before arrival in SI region. The relevant sentence is now modified accordingly.

♣ Page 11, lines 24–27
"When the air parcels travel over the sea, e.g. during 24−18 hours, or 18−6 hours before their arrival in SI, the surface evaporation instantly increases. For instance, between 18 and 6 hours before arrival in SI, the median surface evaporation rate along the trajectories doubles from 0.15 to 0.32 mm h$^{-1}$ with a peak 12 hour before the arrival in SI."

6. Section 4.1.1, third paragraph refers to dots following mixing/Rayleigh lines, but that doesn't really seem to be the case, at least not exactly. Do these lines depend on the constants in their equation, which might fit better for other values? If so, consider using them. If not, or if this is too much effort, explain more clearly how they agree with the scattered dots.
The relevant sentence has been further clarified to describe the scattered dots.

♣ Page 12, lines 10–12
"[…] the upper to low-level trajectories (green to purple dots in Fig. 9a, b) follow a mixing line (dashed line) during their descent while the lowermost trajectories (black and grey dots) are distributed over wider domain and do not follow exactly a Rayleigh distillation line […]"

7. At multiple locations in the text, a moist plume over Africa is mentioned, and it is not always referred to in the same way (African moist plume, African moisture plume, the plume of …), please keep this consistent throughout the manuscript. Also, explain whether this is a climatological plume or a feature of this specific event.
The African moisture plume is often identified when the upper-level trough strengthens and deepens in a south-north orientation over the western Mediterranean. It has been reported in multiple studies, e.g. Chazette et al., 2015, Lee et al., 2015 and 2017. This tip of information has been included in the text, and 'African moisture plume' has been used consistently throughout the manuscript.

♣ Page 15, lines 22–27
"[…] the strong cyclonic flow around the trough (grey dashed line in Fig. 14b) induces the advection of the African moisture plume towards SI and leads to large-scale uplift of the warm and moist air mass along the cold front. The existence of an African moisture plume is often associated with the presence of a deepening, north-south oriented upper-level trough over the western Mediterranean (Chazette et al., 2015; Lee et al., 2016 and 2017).

It brings moisture and leads to gradual rain out of the air parcels over Italy […]"

♣ Page 17, line 24

Chazette, P., Flamant, C., Raut, J.C., Totems, J., and Shang, X.: Tropical moisture enriched storm tracks over the Mediterranean and their link with intense rainfall in the Cevennes-Vivarais area during HyMeX. Q. J. R. Meteorol. Soc., 142, 320–334, doi:10.1002/qj.2674, 2018.

8. Section 5 can still be slightly shortened by removing some of the values (not needed in the conclusions) and considering the text further, specifically the part on p. 15, l. 8–26.
Section 5 has been shortened by removing the values, especially in the second and third paragraphs (p.15, l. 11–29).

**Typos and Formulation**

Page 1, Line 20: replace *linked to a frontal feature* with *ahead of a cold front*
Corrected.

Page 2, Line 4: change to *large amount of convective precipitation*
Corrected.

Page 2, Line 6: replace *preceding the occurrence of* with *before*
Corrected.

Page 2, Line 8: remove comma after SI
Corrected.

Page 2, Line 9: replace *brings* with *lifts*, remove *masses*
Corrected.

Page 2, Line 10: remove *to higher altitudes*
Corrected.

Page 2, Line 23: change *mesoscale deep convection* to *organized deep convection*, add comma after *events*, and remove *along*
Corrected.

Page 2, Line 24: remove *content*, remove *deep*
Corrected.

Page 2, Line 25: replace *question* with *research topic*
Corrected.

Page 2, Line 28-29: Using *either… or* like this indicates that Africa or extratropical remnants of Atlantic tropical cyclones over the Atlantic are the only sources of moisture, but other sources are possible as well as a combination of sources. I suggest to rephrase this a bit to account for these possibilities or at least not exclude them.
Corrected.

♣ From Page 2, line 29
"[…] These studies found substantial contributions of subtropical and tropical moisture coming from various sources such as  Africa (latitude ≥ 20°N)  and the extratropical remnants of Atlantic tropical cyclones, among others."

Page 2, Line 29: Change sentence to *more recent studies (e.g. Lee et al., 2016 and 2017), pointed out a significant moisture contribution, one quarter of the total integrated water vapor, from North Africa…*
Corrected.

Page 3, Line 9: remove the *s* at the end of isotopes
Corrected.

Page 3, Line 11: replace *constraints* with *insights*
Corrected.

Page 3, Line 20: change *at surface* to *at the surface*
Corrected.

Page 4, Line 11: remove comma after *evaporation*
Corrected.

Page 4, Line 15, 17: repetition of *the free troposphere of …*
Removed the one of them.

Page 5, Line 12: change *in the low levels* to either *in the lower troposphere* or simply *at low levels*
Corrected to *in the lower troposphere*.

Page 5, Line 12: what exactly meant by *initiated over Algeria?* Is this where convection first occurred?
Yes, the convection first occurred over Algeria. For sake of clarify, the expression has been revised.

♣ Page 5, lines 14-15
"[…] first occurred over Algeria and […]"

Page 5, Line 13: replace *large* with *high*
Corrected.

Page 5, Line 14: add *s* after *temperature*
Added.
Page 6, Line 5: replace *sense* with *context*
Corrected.

Page 7, Line 15: replace *written as an output* to *extracted from the model* or just change to *written out*
Revised to *written out*.

Page 7, Line 20: insert *specific* before *humidity*, remove comma after *humidity*
Corrected.

Page 8, Line 4: add *s* after *ratio*
Added.

Page 8, Line 10: add *s* after *condition*
Added.

Page 8, Line 17: insert *region* after *SI*
Corrected.

Page 8, Line 23: replace *a reasonable* with *good*
Corrected.

Page 8, Line 25: add *s* after *observation*
Corrected.

Page 9, Line 2-6: The two parts of the sentence *The moisture structure (…) has been further studied…* are extremely separated and the sentence becomes difficult to read. You could rephrase it to something like: *The moisture structure upstream of the HPE studied by Lee et al. (2016) has been further analyzed. Three features are highlighted below: 1) the presence of …*
Corrected.

Page 9, Line 7: add commas before and after *located over south-eastern France*
Added.

Page 9, Line 8: The threshold of 1002 seems arbitrary, this might be a relic from back when the area of MSLP < 1002 hPa was the only shaded part of Fig. 3.
You are right. With the revised figure (better color scale), now the upper-level trough can be defined with a threshold of 1006 hPa.

Page 9, Line 10: remove comma after *air*
Removed.

Page 9, Line 11: remove commas before and after *q*, remove *thus*
Corrected.

Page 9, Line 15-16: change to *(…) is located ahead of the trough (red area in Fig. 3b)*
Corrected.
Page 9, Line 16: change to *Comparing the maps of q and $\delta^{18}O_v$ (crescent enclosed by a dashed line in Fig. 4a-b) reveals an additional…*
Corrected.

Page 9, Line 28, 29: repetition of *the hourly evolution of*
The later one has been removed.

Page 10, Line 3: move *slightly* to after *increase*
Corrected.

Page 10, Line 7: is it the upper level trough or the cold front? Those are two different things.
Corrected.

Page 10, Line 12: remove both, *the* and *level,* at the end of the sentence
Corrected.

Page 10, Line 13: remove *Then* at the beginning of the sentence, add a comma after *UTC*
Corrected.

Page 10, Line 14: which western edge?
It is the western edge of the trough. For sake of clarity, *at the western edge* has been removed.

Page 10, Line 27: insert *at* after *SI*
Corrected.

Page 11, Line 2: insert *the* before *two*
Corrected.

Page 11: Multiple instances of *history* where *evolution* might be the better word (e.g. lines 7 and 11)
We kindly propose to keep 'history' as it gives information about the evolution along the trajectory.

Page 11, Line 10: change title to *Phase one: the convective phase*
Corrected.

Page 11, Line 13-15: change to *The 3-day backward trajectories in Fig. 7 indicate that the air parcels arriving at SI in the 800-700 hPa layer originated over the North Atlantic.*
Corrected.

Page 11, Line 16: replace *are mostly* with *remain*
Corrected.

Page 11, Line 17: add *comma* after *SI, remove* rapidly
Corrected.

Page 11, Line 20: change *as well as* to *and*
Corrected.

Page 11, Line 22: change *mixing* to *they mix*
Corrected.

Page 11, Line 23: remove *occurs,* change *the median q value* to *the median of q*
Corrected.

Page 11, Line 27: change to *(…) before their arrival, showing that q and $\delta^{18}O_v$ increase rapidly in the last 12 hours before the parcels arrive over SI. Between 60 and 12 hours before their arrival (Fig. 8a, b), q and $\delta^{18}O_v$ are still relatively small, at around 2-6 g kg$^{-1}$ and between -25 and -19 ‰, respectively. (Is there really in the dry pocket of upper-level trough? The trough itself should also be somewhere else this long before the event…)*
Corrected.

Page 12, Line 3: change to *for conditions in the Mediterranean*
Corrected.

Page 12, Line 3: *blue line* I do not see a blue line in the figure, does this refer to the dashed line?
Yes. It has been corrected to dashed line.

Page 12, Line 6: gray to purple – shouldn't this be green to purple?
Thanks for pointing this out. Green to purple is correct.

Page 12, Line 14: change to *Horizontal SWI distribution*
Corrected.

Page 12, Line 15: remove *feature,* remove *a region of enhanced convective activity and*
Corrected.

Page 12, Line 16-18: change *multiple convective cells* to *a convective line, which extends (…) and is located ahead of the surface cold front. Westerly and north-westerly winds prevail at 542 m ASL while south-westerly wind is dominant at 2455 m ASL.*
Corrected.

Page 12, Line 18-21: change to *Within the precipitation area, lower $\delta^{18}O_v$ values (< -16‰) than in the vicinity are found at 542 m ASL while locally higher $\delta^{18}O_v$ values are found at 2455 m and 5565 m ASL (Fig. 10d, f), indicating the presence of strong and deep convective mixing. This convection causes the vertical transport of SWI-depleted moisture…*
Corrected.

Page 13, Line 1: remove *large*
Corrected.

Page 13, Line 7: remove *of*
Corrected.

Page 13, Line 9: replace *at* with *along*
Corrected.

Page 13, Line 10: change to *Additional moisture is then…*
Corrected.

Page 13, Line 12: add *s* after *hour*, remove *the… values*
Corrected.

Page 13, Line 13: change *strongly increase* to *increase strongly*
Corrected.

Page 13, Line 19: remove third zero
Removed.

Page 13, Line 23: add comma after *altitudes*
Corrected.

Page 13, Line 24: the vapor take-up isn't over North Algeria, it's over the Mediterranean sea
Corrected to water *vapour in the Strait of Sicily*.

Page 13, Line 24: end sentence after *(Fig. 11a-b)* and remove *and* before *the median q,* change to *the median of q*
Corrected.

Page 13, Line 25-26: change *precipitation onset* to *the onset of precipitation*
Corrected.

Page 13, Line 26, 27: repetition of *moist and SWI-enriched*
Corrected.

Page 14, Line 2: add *s* at the end of *parcel,* add comma after *particular*
Added.

Page 14, Line 7: change to *Horizontal SWI distribution*
Corrected.

Page 14, Line 12: this is still hardly visible at 5500 m, all shades of blue look almost exactly the same in print
For sake of clarity, the values have been stated in the sentence.

♣ Page 14, lines 15-17
"[…] and $\delta^{18}O_v$ are relatively low from near the surface (between −22 and −26 ‰, 542 m ASL) to mid altitudes of 5565 m ASL (between −30 and −36 ‰) (Fig. 13b, d, and f)."

Page 14, Line 15: the black line shows very little decrease, it's visible for the others though multiple repetitions of *moisture plume over North Africa* or similar in this line.
Corrected to 'red to purple line' and the repetitions have been corrected.

Page 14, Line 23: replace *becomes more depleted* with *decreases*
Corrected.

Page 14, Line 24: replace *that* with *which*
Replaced.

Page 14, Line 24: I thought P2 was dominated by large scale ascent, not convection.
The sentence has been clarified.

♣ Page 14, line 27-28
"[…] the moistures which feeds the convection during P2 is related to large scale ascent from North Africa"

Page 14, Line 24-28: This paragraph seems like it belongs to section 4.2.1, not 4.2.2.
As this paragraph is a summary of Section 4.2, we kindly propose to keep the location at the end of the section.